# Current global estimates, risk factors, and knowledge gaps for Hepatitis E virus (HEV): A scoping review

Md Koushik Ahmed[1], Hanna Maroofi[2], Madeleine Blunt[3], Alain Labrique[4], Carl Kirkwood[5], Kirsten Vannice[5], Kawsar R. Talaat[3], Julia Lynch[6], Brittany L. Kmush[1]*

1 Department of Public Health, Syracuse University, Syracuse, New York, United States of America,
2 Department of Epidemiology, Johns Hopkins Bloomberg School of Public Health, Baltimore, Maryland, United States of America, 3 Department of International Health, Johns Hopkins Bloomberg School of Public Health, Baltimore, Maryland, United States of America, 4 Department of Digital Health and Innovation, World Health Organization, Geneva, Switzerland, 5 The Gates Foundation, Seattle, Washington, United States of America, 6 Office of the Director General, International Vaccine Institute, Seoul, South Korea

* blkmush@syr.edu

## Abstract

Hepatitis E virus (HEV) remains a leading cause of acute viral hepatitis globally, particularly in South Asia and Africa. However, epidemiological prioritization is hampered by fragmented data and discordant disease burden estimates. Following JBI and PRISMA-Sc guidelines, we conducted a scoping review of global HEV evidence. We used the PCC framework: (P) general and high-risk populations (pregnant women, immunocompromised, and displaced groups); (C) quantitative estimates of burden, risk factors, or virological gaps; and (C) global evidence across all WHO regions to include studies. We searched PubMed, Scopus, and Web of Science, supplemented by country-specific searches in Google Scholar and IHME. From 11,583 citations, 395 articles met the inclusion criteria. The temporal distribution shows a marked increase in research volume, with 65.3% of studies published after 2010; however, 54.9% relied on observational descriptive designs while experimental investigations remained infrequent (4.3%). We identified three estimates of the global burden of HEV: the IHME Global Burden of Disease (GBD) published in 2021 (19.4 million cases) and two widely cited systematic reviews published in 2012 (20.1 million infections) and 2020 (939 million infections). A significant virological "blind spot" was observed, as 47.8% of studies did not report genotype information, though Genotype 3 (21.8%) was the most frequently identified among specified reports. Key risk domains identified were environmental (sanitation/water contamination) and cultural/occupational practices. Pregnant women, immunocompromised patients, and patients with pre-existing liver conditions were high at-risk populations. Key knowledge gaps identified were limited confidence in burden of disease estimates: severe molecular blind spots and evidence deserts, limited public health resources for surveillance, diagnostics, and reporting of cases and deaths in highest risk settings; exclusion of outbreaks from estimates of the

**Data availability statement:** All relevant data are within the manuscript and its Supporting Information files. The minimal data set required to replicate all study findings—including the complete evidence synthesis extraction table and country-level burden data—is provided in S1–S3 Tables.

**Funding:** MKA, HM, AL, KRT, JL, BLK were supported by the Gates Foundation (Investment ID INV 016643). KV, CK received a salary from the Gates Foundation. The funder commissioned the study but had no role study design, data collection and analysis, and decision to publish.

**Competing interests:** The authors have declared that no competing interests exist.

burden of disease and unreliable convenience sample derived estimates. Hepatitis E virus is often neglected by international communities, global actors and national governments. However, it is difficult for stakeholders to prioritize a pathogen with highly variable and unreliable global burden of disease estimates. Comprehensive country level data based on more access to routine testing could facilitate global initiatives to devise strategies for equitable vaccination and mitigate the morbidity and mortality associated with this vaccine-preventable disease.

## Author summary

Hepatitis E is a major cause of acute hepatitis particularly in South Asia and Africa. Fatality rates are high for pregnant women and patients with pre-existing conditions. However, there is a lack of consensus about the global burden of HEV disease. Global and country-level estimates often vary dramatically. In this scoping review, we aim to summarize the latest evidence and estimates to understand the knowledge gaps related to hepatitis E global burden estimates and risk factors. From the available studies, we extracted genotype, anti-HEV seropositivity, reported outbreaks and the year of most recent outbreak reported in the country specific studies, and HEV incidence for each country. We also tried to identify and confirm specific missing data points for each country. Our scoping review found that there is a severe lack of data on HEV incidence and mortality for many countries across WHO regions. We found a wide range of variations of the global estimates across the countries and populations. Comprehensive country level data based on more access to routine testing could facilitate global initiatives to devise strategies for equitable vaccination and mitigate the morbidity and mortality associated with this vaccine-preventable disease.

## Introduction

Hepatitis E virus (HEV), a member of *Hepeviridae* family [1], was first called enterically-transmitted non-A, non-B hepatitis. While not isolated until the 1980's, it has been a recognized cause of large-scale outbreaks in Southeast Asia since the 1950s [2]. Since then, it has emerged as one of the most important causes of acute hepatitis in both developing and developed countries [3,4]. The HEV species that causes human disease has eight genotypes [5]. HEV genotypes 1 and 2, which are largely transmitted through contaminated water, are responsible for most human infections and are of concern for large-scale outbreaks. Genotypes 3 and 4 also cause infections in humans but are zoonotic and are found in several animal species, notably wild and domestic swine [6]. Genotypes 5 and 6 have been found only in wild boar and genotypes 7 and 8 in camels [7]. HEV genotype 1 and 2 are considered endemic in South Asia and Africa [8] while HEV genotype 3 and 4 are mainly observed in the United States, Europe, and East Asia [9].

Hepatitis E (HE) disease presents as acute, viral hepatitis, including jaundice, abdominal pain, fever, fatigue, and anorexia [10]. There is no specific treatment except for supportive care. In the general population, the disease is usually mild and self-limiting with only a 0.1%–4% case fatality rate [11] though chronic infections leading to cirrhosis have increasingly been recognized in immunocompromised individuals such as organ transplant recipients [9]. However, certain populations are prone to severe disease, most notably pregnant women. Pregnant women with HE are more likely to have fulminant hepatic failure and intrapartum hemorrhage, with case fatality rates ranging from 10% to 40% during pregnancy [12–15]. Children are often less likely to experience clinical symptoms of HE, although there have been outbreaks where children are a major impacted group [16]. In sharp contrast to Hepatitis A, children also generally have lower rates of seropositivity [17,18]. HEV causes large scale outbreaks but is also responsible for large proportions of acute viral hepatitis cases presenting to hospitals in endemic areas [19,20]. Most cases in endemic areas are adolescents and young adults. As of now, two vaccine candidates have undergone clinical trials [21,22]. Only one, Hecolin, completed clinical trials and is now licensed in China (since 2011) and Pakistan (since 2015) for use in healthy adults. This recombinant vaccine was found to be 93% effective for preventing clinical disease after four years of follow up [23,24] and 87.3% effective after 10-years of follow up [25]. However, the vaccine has not been submitted for pre-qualification by the WHO and is not widely used.

Despite these clinical and preventative advancements, global public health prioritization of HEV is hindered by a critical lack of consensus on its true epidemiological scale. This is evidenced by the staggering discordance between major global burden estimates—ranging from approximately 20 million to nearly 1 billion infections [20,23,25]. Such fragmentation suggests that current literature presents a complex landscape of risk factors and prevalence data that have not been systematically synthesized to identify overarching global trends. Therefore, a scoping review is the most appropriate methodology for this problem; unlike a narrow systematic review, this approach allows for the comprehensive mapping of a heterogeneous evidence base to identify specific geographic "Evidence Deserts" and "Molecular Blind Spots" that currently obstruct evidence-based policy and equitable vaccine deployment. A scoping review is therefore necessary to systematically map this heterogeneous evidence base, clarify the limitations of current burden estimates, and identify the specific areas where data is missing. This mapping is a critical prerequisite for guiding future research priorities and informing global public health policy, particularly regarding vaccine deployment and intervention strategies in high-risk populations.

## Methods

This scoping review followed the framework of Joanna Briggs Institute (JBI) [26] and the Preferred Reporting Items for Systematic Reviews and Meta-Analyses extension for Scoping Reviews (PRISMA-ScR) [27]. In accordance with JBI guidelines, which define the consultation phase (Step 6) as optional, this review focused on the systematic identification, selection, and synthesis of existing evidence (Steps 1–5) to map the global landscape of HEV epidemiology. The protocol was registered in the Open Science Framework (https://doi.org/10.17605/OSF.IO/A4CV7). A completed PRISMA-ScR checklist is provided in S1 Checklist to ensure reporting transparency.

### Step 1: Identify the research question

The primary objective of this review was to map the global landscape of Hepatitis E Virus (HEV) epidemiology and identify critical gaps in the evidence base. We hypothesized that current burden estimates are characterized by significant geographical and methodological heterogeneity. The specific research questions were:

1. What are the current global and country-specific estimates of HEV disease burden (incidence and seroprevalence)?

2. What are the documented risk factors for HEV infection?

3. Which populations (e.g., pregnant women, immunocompromised, or displaced persons) are at the highest risk for severe clinical disease?

4. What are the primary knowledge gaps regarding genotype distribution and burden estimates?

**Step 2: Identify the relevant literature**

We employed a three-tiered search approach to ensure comprehensiveness across indexed and non-indexed literature:

1. **Systematic Search:** We searched PubMed, Scopus, and Web of Science for studies published through December 30, 2023. The search strategy (Table 1) used keywords and MeSH terms organized into three concepts: HEV estimates, risk factors, and vulnerable populations. Full search syntaxes for all databases are available in S1 File. We conducted systematic search at two different times to make sure that the search syntax offers consistent and comprehensive results. We conducted the final systematic search during the week of Jan 5–9, 2024, and exported the bibliographic citations for screening, full text review and data extraction.

2. **Purposive Country-Specific Search:** To mitigate indexing bias, we conducted searches in Google Scholar for each country (e.g., "Hepatitis E Bangladesh") following the World Health Organization regions in Jan 2024. In order to ensure comprehensiveness and report on the most recent published studies, we also conducted final round of purposive country-specific search through Google Scholar in July 2024. These studies informed most recent country specific data for country-specific HEV genotype, anti-HEV seropositivity, reporting of recent outbreaks and year of most recent outbreaks. To ensure reproducibility, we screened the first 50 results for each query. To capture data heterogeneity, when multiple studies reported differing anti-HEV seroprevalence for the same country, we extracted both the highest and lowest reported estimates.

3. **Grey Literature & Global Databases:** We searched authoritative sources, including WHO regional reports, strategy documents, guidelines and other grey literature which had relevant data points for our review questions. We searched for a country specific IHME incidence data of acute hepatitis E in December 2024 [28].

While the search was restricted to English-language publications due to resource constraints for high-quality translation, international surveillance reports were utilized to maximize global coverage.

**Step 3: Select the literature**

Citations were managed in Rayyan [29]. Following duplicate removal, a two-stage screening process (title/abstract followed by full-text) was conducted by three independent reviewers (MKA, HM, MB). Discrepancies were resolved through

**Table 1. Full executable search string by database.**

| Database | Full Executable Search String |
|---|---|
| PubMed | ("hepatitis e estimates"[title/abstract:~3] OR "hepatitis e update"[title/abstract:~3] OR "hepatitis e seroprevalence"[title/abstract:~3] OR "Hepatitis E"[Mesh] OR "Hepatitis E virus"[Mesh] OR "ORF3 protein, Hepatitis E virus"[Supplementary Concept] OR "ORF2 protein, Hepatitis E virus"[Supplementary Concept] OR "Hepevirus"[Mesh] OR "Hepatitis e risk factors"[title/abstract:~3] OR "risk factors"[MeSH Terms]) AND (english[Filter]) |
| Scopus | (TITLE-ABS-KEY("Hepatitis E" OR "HEV" OR "Hepevirus") W/5 (estimate* OR update* OR seroprevalen* OR epidemiol* OR status OR synopsis OR "risk factor*" OR "vulnerable population*" OR "knowledge gap*")) AND (LIMIT-TO (LANGUAGE, "English")) |
| Web of Science | (TS=("Hepatitis E" OR "HEV" OR "Hepevirus") NEAR/5 (estimate* OR update* OR seroprevalen* OR epidemiol* OR status OR synopsis OR "risk factor*" OR "vulnerable population*" OR "knowledge gap*")) AND LA=(English) |

consensus or by the Principal Investigator (BLK). We applied the **PCC (Population, Concept, and Context)** framework to ensure a comprehensive mapping of the HEV research landscape:

1. **Population (P):** The review included studies focusing on the general population as well as specific high-risk sub-groups, specifically: pregnant women, immunocompromised patients, occupational risk groups, and displaced or humanitarian populations.

2. **Concept (C):** The primary concept was the characterization of evidence on disease burden and global knowledge gaps. This was operationalized by evaluating three interrelated domains:

   a. **Epidemiological Burden:** Global and country-specific estimates of incidence and seroprevalence.

   b. **Risk Determinants:** Documented environmental, occupational, and population-specific factors (e.g., severe disease in pregnancy).

   c. **Virological Gaps:** Completeness of genotype distribution reporting and its alignment with disease burden estimates.

3. **Context (C):** Global evidence across all six WHO regions.

We included original research articles of all study designs, as well as peer-reviewed evidence syntheses (including systematic, meta-analytic, and comprehensive literature reviews) that provided regional or global HEV burden estimates. Modeling studies, such as those from the Institute for Health Metrics and Evaluation (IHME), were also included to provide a benchmark for clinical incidence and mortality data.

Exclusion Criteria**:** We excluded brief editorials, commentaries, and non-systematic opinion pieces and non-authoritative grey literature (e.g., news blogs) that did not provide primary data or formal epidemiological estimates. We also excluded other scoping reviews to avoid duplicative reporting.

### Step 4: Chart the data

The *a priori* structured matrix using Microsoft Excel spreadsheet was developed to extract data from the included studies. The review matrix included study objective, year of publication, study design, country, WHO region, populations, HEV genotypes, reported risk factors, seroprevalence, seroprevalence estimate year, reported outbreak (yes/no), number of cases and gaps identified in the study. Data extraction was performed independently by reviewers (MKA, HM, MB) and pilot-tested on 10 studies to ensure consistency. The data extraction by other reviewers was further reviewed by the author. To address the significant heterogeneity in how study designs were reported across the 395 included articles, we developed a systematic categorization framework. We employed a hierarchical string-matching taxonomy to collapse diverse study descriptions into five functional categories based on their primary epidemiological contribution:

- **Evidence Synthesis**: Systematic reviews, meta-analyses, and comprehensive literature reviews. These were utilized to provide aggregated burden estimates and to identify historical trends in regional HEV epidemiology.

- **Surveillance/Public Health**: National microbiological surveys, blood donor screening programs, and outbreak investigations used to track transmission.

- **Observational – Analytical**: Studies employing comparison groups or longitudinal follow-up, including case-control, cohort, and post-hoc analyses.

- **Experimental/Laboratory-based**: In vitro experiments, vaccine trials, and laboratory-based diagnostic validations.

- **Observational – Descriptive**: The majority of studies, encompassing cross-sectional seroprevalence surveys, case reports, and molecular/phylogenetic analyses used for genotype characterization.

Simultaneously, a country-specific matrix following the six WHO regions was used to capture human genotypes, anti-HEV seropositivity ranges, and the timing of outbreaks. The extracted data files can be accessed in S1–S3 Tables.

**Step 5: Collate, summarize, and report results**

We employed a narrative synthesis approach combined with quantitative descriptive mapping to integrate the findings from 395 included studies. Results were organized and reported in three main stages to provide a comprehensive overview of the HEV landscape:

1. **Descriptive analysis of studies:** We used descriptive statistics (frequency counts and percentages) to characterize our included studies. Specifically, we quantified the distribution of studies by publication period, study design, WHO region, HEV genotype, study population (e.g., general population vs. high-risk groups like maternal and displaced populations) and reported risk factors by thematic synthesis.

2. **Global, regional and country-specific analysis:** We synthesized data regarding the global burden of HEV-caused cases and deaths. By comparing IHME-modeled incidence [28] with localized clinical and serological data, we established the current evidence base and epidemiological estimates of hepatitis E. Findings were categorized by overview of global estimates and the six WHO regions (AFRO, EURO, SEARO, WPRO, AMRO, and EMRO). For each region, evidence was summarized in standardized tables reporting country-specific genotypes, seropositivity ranges, IHME-reported incidence, the year of the most recently documented outbreak and the study references.

3. **Risk factors and gap analysis:** We applied a qualitative coding framework to the 395 included studies to synthesize reported risk factors and vulnerable population data. Specific risk factors were extracted, synthesized into five thematic domains and quantified by the frequency of each specific factor using a bubble plot. For gap analysis we employed a two-stage analytical approach to quantify the gap between disease burden and evidence base for each country:

   1. **Calculating the Evidence Maturity Index (EMI):** To quantify the depth and breadth of evidence availability, we applied Evidence Maturity Index (EMI). This metric was adapted from the 'Data Readiness and Maturity' frameworks described and applied in other studies [30,31]. By assigning hierarchical weights (W = 0 to W = 3) to represent the transition from absolute data voids to evidence maturity, we normalized regional research intensity to identify 'Data Deserts' where predicted HEV burden significantly outstrips evidence output. Countries were assigned weights (W) based on the depth of available data:

      - Absolute Desert (W=0): No HEV-specific literature identified.

      - Epidemiological Desert (W=1): High predicted burden ($>10,000 cases) but zero clinical records.

      - Virological Desert (W=2): Clinical presence documented, but local genotype data is missing.

      - Evidence Mature (W=3): Clinical, serological, and molecular (genotype) data are all present.

The EMI was calculated as a normalized weighted average:

$$EMI = \sum (\% \text{ Category} * W)/3 * 100$$

   2. **Regional Discordance Mapping:** The EMI was utilized as the "Knowledge Base" (X-axis) and plotted against the $Log_{10}$ estimated annual incidence ("Disease Burden," Y-axis). Following the methodology used in other studies [32,33], we established a "Balanced Path" using a reference line *(y = 1.0 + 0.08x)* representing the expected research-to-burden equilibrium. This enabled the identification of Priority Blind Spots—high-burden countries where the evidence base is critically insufficient to guide precision public health interventions.

This analytical framework allowed us to not only map the burden, but also highlight the gap related to HEV burden, identifying the 'Priority Blind Spots' where the absence of evidence most critically hinders precision public health intervention. All analysis were performed using R (version 4.5.2).

## Results

### Characteristics of included studies

We found 11,583 bibliographic citations to undergo title and abstract screening from systematic database searches (Fig 1).

A total of **395 studies** were included in the final synthesis (Table 2). The temporal distribution shows a marked increase in HEV research over time, with 258 studies (65.3%) published between 2010 and 2024, compared with only 30 studies (7.6%) published prior to 1995. Most studies employed observational descriptive designs (217, 54.9%), followed by surveillance and public health investigations (72, 18.2%). Analytical observational studies accounted for 45 (11.4%) of

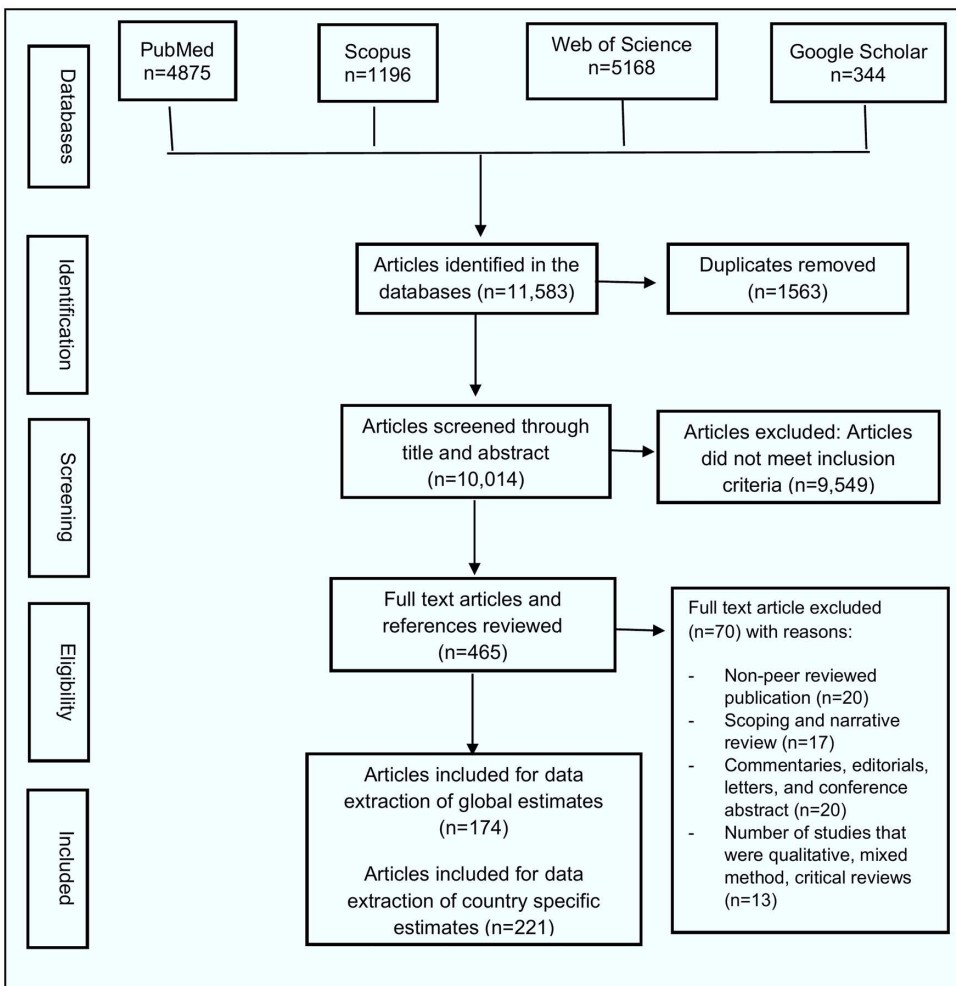

**Fig 1. Flow diagram of study selection.** A PRISMA-compliant flow chart documenting the identification, screening, and inclusion process. From 11,583 initial citations, 174 met criteria via database search, with an additional 221 identified through country-specific purposive searching, resulting in a total of 395 included studies.

**Table 2. Characteristics of included studies (N = 395).**

| Characteristics | Frequency (n, %) |
|---|---|
| **Year of publication** | |
| 1950–1964 | 1 (0.3) |
| 1965–1979 | 1 (0.3) |
| 1980–1994 | 28 (7.1) |
| 1995–2009 | 107 (27.1) |
| 2010–2024 | 258 (65.3) |
| **Study Design** | |
| Observational – Descriptive | 217 (54.9) |
| Surveillance/ Public Health | 72 (18.2) |
| Observational – Analytical | 45 (11.4) |
| Secondary/Synthesis | 44 (11.1) |
| Experimental/Laboratory-based | 17 (4.3) |
| **WHO Region** | |
| AFRO | 103 (26.1) |
| EURO | 100 (25.3) |
| SEARO | 61 (15.4) |
| WPRO | 48 (12.2) |
| AMRO | 42 (10.6) |
| EMRO | 33 (8.4) |
| Multi-region | 8 (2.0) |
| **HEV Genotype** | |
| Genotype 1 (G1) | 54 (13.7) |
| Genotype 2 (G2) | 4 (1.0) |
| Genotype 3 (G3) | 86 (21.8) |
| Genotype 4 (G4) | 8 (2.0) |
| Genotype 7 (G7) | 1 (0.3) |
| Multiple Genotypes (Mixed) | 53 (13.4) |
| Not reported/specified | 189 (47.8) |
| **Study Population** | |
| General Population | 98 (24.8) |
| Symptomatic/Acute Hepatitis Patients | 63 (15.9) |
| Maternal & Neonatal | 48 (12.2) |
| Occupational & Zoonotic Risk Groups | 42 (10.6) |
| Clinical High-Risk (Immunocompromised/CLD) | 39 (9.9) |
| Other/Mixed Groups | 37 (9.4) |
| Displaced & Humanitarian | 32 (8.1) |
| Blood & Organ Donors | 29 (7.3) |
| Animal & Environmental Sources | 7 (1.8) |
| **HEV Risk Factors** | |
| **Water/Env** | |
| Contaminated Water | 178 (45.1%) |
| River/Surface Water | 31 (7.8%) |
| Floods/Monsoon | 15 (3.8%) |
| **Food/Zoonotic** | |
| Pork Products | 102 (25.8%) |
| Raw/Undercooked Meat | 58 (14.7%) |

*(Continued)*

**Table 2.** (Continued)

| Characteristics | Frequency (n, %) |
| --- | --- |
| Occupational Swine Contact | 49 (12.4%) |
| Wild Game/Bushmeat | 21 (5.3%) |
| **Sanitation** | |
| Poor Sanitation | 88 (22.3%) |
| Sewage Contamination | 44 (11.1%) |
| **Clinical/Host** | |
| Older Age | 76 (19.2%) |
| Immunosuppression | 42 (10.6%) |
| Male Sex | 31 (7.8%) |
| Pregnancy | 26 (6.6%) |
| **Socio-Demo-Cultural** | |
| Displacement/Camps | 44 (11.1%) |
| Rural Residence | 39 (9.9%) |
| Travel History | 25 (6.3%) |

included reports, while secondary or synthesis studies comprised 44 (11.1%). Experimental or laboratory-based investigations were relatively infrequent (17, 4.3%).

**Geographic and genotypic distribution.** Studies were geographically diverse but unevenly distributed across WHO regions. The largest contributions originated from AFRO (103, 26.1%) and EURO (100, 25.3%), followed by SEARO (61, 15.4%), WPRO (48, 12.2%), AMRO (42, 10.6%), and EMRO (33, 8.4%). Multi-regional studies were uncommon (8, 2.0%).

HEV genotype was specified in 206 studies (52.2%), with Genotype 3 reported most frequently (86, 21.8%), followed by Genotype 1 (54, 13.7%). Mixed-genotype infections were identified in 53 studies (13.4%). Nearly half of all studies (189, 47.8%) did not report genotype information.

**Study populations.** The general population was the most commonly studied group (98, 24.8%), followed by symptomatic or acute hepatitis patients (63, 15.9%). Maternal and neonatal populations comprised 48 studies (12.2%), while occupational and zoonotic risk groups accounted for 42 (10.6%). Clinical high-risk populations, including immunocompromised individuals and those with chronic liver disease, were examined in 39 studies (9.9%). Studies conducted in displaced or humanitarian settings represented 32 (8.1%) of the total.

**Reported HEV risk factors.** HEV risk factors were reported across multiple, non-mutually exclusive domains. Environmental and water-related exposures were most frequently identified, with contaminated water reported in 178 studies (45.1%), followed by poor sanitation (88, 22.3%) and sewage contamination (44, 11.1%).

Foodborne and zoonotic exposures were commonly reported, particularly pork product consumption (102, 25.8%), raw or undercooked meat (58, 14.7%), and occupational swine contact (49, 12.4%).

Among host-related factors, older age was most frequently reported (76, 19.2%), followed by immunosuppression (42, 10.6%), male sex (31, 7.8%), and pregnancy (26, 6.6%). Socio-demographic factors included displacement or residence in camps (44, 11.1%), rural residence (39, 9.9%), and travel history (25, 6.3%).

Below, we summarize the data relating to our specific research questions.

### What are the current global and country specific estimates of Hepatitis E disease burden?

**Overview of global estimates.** There are a wide range of estimates of the mortality and burden of disease caused by HEV. Table 3 highlights the 4 widely documented sources of global estimates. According to Institute for Health Metrics and Evaluation (IHME) data, there were only 3450 deaths from HEV across the globe in 2021 [32]. However, another

**Table 3. Estimates of the Global burden of HEV-Caused Cases and Deaths for data coverage years.**

| Source | No. infections | No. cases | No. deaths | Data coverage years | Reference year | Publication year |
|---|---|---|---|---|---|---|
| Rein et. al. [34] | 20.1 million (95% CI.: 2.8-37.0) | 3.4 million (95% CI.: 0.5-6.5) | 70,000 (95% CI.: 12,400–132,732) | 1990-2010 | 2010 | 2012 |
| IHME [35] | Not reported | 19.5 million (95 CI: 16.0-23.4) | 1930 (95% CI: 1120–2880) | 1990-2019 | 2019 | 2020 |
| Li et. al. [36] | 939 million | Not estimated | Not estimated | 1993-2019 | 2019 | 2020 |
| IHME [37] | Not reported | 19.4 million (95 CI: (16·1- 23·2) | 3450 (95% CI: 1730–5900) | 1990-2021 | 2021 | 2024 |

commonly cited systematic review estimated that there were 70,000 deaths from HEV in genotype 1 and 2 areas in 2005, including 3,000 stillbirths [34].

As pregnant women are at the greatest risk of severe consequences from HEV infection, several studies have estimated the number of deaths caused by HEV during pregnancy. One study from Bangladesh, found that approximately 10% of all maternal mortality is likely due to HEV, which suggests HEV leads to 10,500 pregnancy deaths per year in Southeast Asia [38]. However, another group found that up to 25% of maternal deaths are caused by HEV, which suggests 27,000 deaths are due to HEV each year [39]. A prospective study found a maternal or neonatal HEV death rate of 2.9 per 1000 pregnancies (1.2 per 1000 maternal deaths and 1.7 per 1000 neonatal deaths), although this was an analysis of data collected in the 1990s [40]. This estimate would indicate that nearly 60,000 deaths in Southeast Asia are caused by HEV per year. However, these are all indirect estimates of the total burden of mortality from HEV as diagnostic testing and reporting of the deaths is limited.

The number of HE cases and HEV infections is also difficult to estimate due to poor surveillance and lack of reporting. IHME estimates that there were 19.4 million cases of HE in 2021 [32]. Another widely cited source estimated that there were 20.1 million infections with 3.4 million of those being symptomatic infections in Southeast Asia and Africa during 2005 [34]. A recent systematic review and meta-analysis estimated that there are approximately 110 million infections of HE each year [36].

A systematic review that examined global evidence from 1978 to 2015 found reported HEV outbreaks in 12 countries in Asia, 14 countries in Africa, 2 in Europe, and 3 in North America [41]. However, systematic reviews on the global burden of HEV often do not incorporate estimates from outbreak investigations in displaced persons camps due to the jurisdictional ambiguity as it relates to country-wide estimates. Conflict and displacement are often associated with the conditions that support HEV outbreaks [41]. A large portion of global HEV outbreaks occur in Africa [42], and about 50% of 20 outbreaks across 9 Sub Saharan African countries were reported to occur in camps of refugees and internally displaced persons (IDPs) in countries with significant warfare, conflicts and human displacements: Kenya (1702 cases) [43,44]; South Sudan (>5000 cases) [45]; Angolan, Sudanese and Somalian refugees in Namibia [46]; Chad (>900 cases) [47,48]; Darfur, Sudan (2621 cases) [49,50]; Uganda (144 cases) [51] and Nigeria (146 cases) [52]. Furthermore, investigation of infection sources is especially challenging in these settings. A study on HEV in Sub Saharan Africa demonstrates that only 3 of 20 reported outbreaks from 9 countries were investigated for their sources of infection [42].

Several limitations associated with these various global estimates of infections include lack of and poor quality of country level data from LMICs. For example, more sources of HEV data from European countries were available for IHME, compared to very few sources from the countries of Asia and Africa where the burden of HEV is highest [4,53] (Table 4). There are only 8 sources from Africa and 34 from Asia, yet most of those are from East and Central Asia. There are 28 sources from European countries despite a smaller burden of disease in Europe compared to Southeast Asia and Africa (Table 4).

**Table 4. Serological and epidemiological data sources informing IHME's 2021 Hepatitis E prevalence estimates.**

| Country | Number of Sources | Year | Testing Kit Brand | Source Population |
|---|---|---|---|---|
| WHO Africa Region (n = 8) | | | | |
| Burundi | 1 | 1997 | Abbott | Urban adults |
| Gabon | 1 | 2008 | Genelabs | Pregnant women |
| Ghana | 1 | 1998 | *unspecified* | Children |
| Sierra Leone | 1 | 1998 | unspecified | Primary school children |
| South Africa | 1 | 1996 | Abbott | Urban and rural black South African adults living in formal housing, squatter camps, or mud huts |
| Tanzania | 2 | 1998 | Abbott | General adult population |
| Tanzania | 1 | 2000 | Abbott | Women |
| WHO Americas Region (n = 12) | | | | |
| Argentina | 1 | 1997 | Abbott | Blood donors, children, hospitalized patients |
| Bolivia | 1 | 1999 | unspecified | Blood donors |
| Brazil | 1 | 1997 | Abbott | Blood donors, patients with acute viral hepatitis, hemodialysis patients, & carriers of schistosomiasis |
| Chile | 1 | 1994 | Abbott | Healthy children, male prisoners, and blood donors |
| Chile | 1 | 1996 | unspecified | Alcoholics, hemophiliacs, blood donors, subjects with acute hepatitis |
| Chile | 1 | 1997 | unspecified | Blood donors, health care workers, inmates in state prisons |
| Cuba | 1 | 2010 | Genelabs | Individuals 5–60 without history of Jaundice |
| Mexico | 1 | 1996 | Abbott | General population |
| Mexico | 1 | 1998 | unspecified | Pregnant women |
| Uruguay | 1 | 1997 | Abbott | Patients at a clinic |
| Venezuela | 1 | 1994 | Abbott | Pregnant women |
| United States | 1 | 1998 | unspecified | Pregnant women |
| WHO's Eastern Mediterranean Region (n = 9) | | | | |
| Israel | 1 | 1995 | In-house | Healthy subjects |
| Iran | 1 | 2003 | Dia.pro | General population |
| Iran | 1 | 2009 | Dia.pro | General population |
| Iran | 1 | 2009 | Dia.pro | General population |
| Iran | 1 | 2012 | Dia.pro | General population |
| Saudi Arabia | 1 | 1994 | unspecified | General population |
| Lebanon | 1 | 1998 | unspecified | Blood donors |
| Yemen | 1 | 1999 | Genelabs | Villagers and domestic animals |
| Tunisia | 1 | 2011 | MP Diagnostics | Blood donors and acute hepatitis patients |
| WHO European Region (n = 33) | | | | |
| Albania | 1 | 2001 | not specified | General population |
| Belgium | 1 | 2012 | not specified | Gynecological (mainly fertility center) or Orthopedic clinics |
| France | 1 | 2007 | not specified | Blood donors |
| Greece | 1 | 1996 | Abbott | Hemodialysis patients and healthy volunteers |
| Greece | 1 | 1998 | Abbott | Blood donors, refugees from southern Albania, children, injection drug users, and at risk patients |
| Italy | 1 | 1996 | Abbott | General population, including drug users and patients of chronic hemodialysis |
| Italy | 1 | 1996 | unspecified | General population |
| Italy | 1 | 1997 | Abbott | Children |

*(Continued)*

**Table 4.** (Continued)

| Country | Number of Sources | Year | Testing Kit Brand | Source Population |
|---|---|---|---|---|
| Italy | 1 | 1998 | unspecified | unspecified |
| Italy | 1 | 1998 | Abbott | Blood donors and in healthy persons in Calabria |
| Italy | 1 | 2003 | Abbott | General population, intravenous drug users, hemodialysis patients |
| Italy | 1 | 2007 | unspecified | General population and workers at zoonotic risk |
| Moldova | 1 | 1997-1998 | In-house | Swine farm workers |
| Netherlands | 1 | 1992 | Genelabs | Low risk blood donors |
| Netherlands | 1 | 1995 | Abbott | Hemophiliacs, blood donors, and hepatitis patients |
| Netherlands | 1 | 2005 | Abbot and Genelabs | Patients with clinical signs of hepatitis |
| Netherlands | 1 | 2006-2007 | not specified | Non-Western immigrants & municipalities with low immunization coverage |
| Netherlands | 1 | 2007 | Genelabs | Blood donors and acute hepatitis patients |
| Portugal | 1 | 1998 | Abbot | Blood donors and chronic liver disease patients |
| Russia (and Belarus) | 1 | 1997 | Abbott | HIV-infected individuals and AIDS patients |
| San Marino | 1 | 1990-1991 | Abbott | Adults 20–79 years old |
| Spain | 1 | 1995 | Abbott | Healthy pregnant women, Moroccan subjects, blood donors, children, and intravenous drug users |
| Spain | 1 | 1996 | Abbott | Children |
| Spain | 1 | 1999 | Abbott | Blood donors, hemodialysis patients, and children infected post transfusion with hepatitis C |
| Spain | 1 | 2004 | Abbott | Pregnant Women |
| Spain | 1 | 2006 | unspecified | Adults from 15-74 years old |
| Spain | 1 | 2007 | Abbot | Health card database |
| Switzerland | 1 | 1994 | Abbott | General population |
| Turkey | 1 | 1993 | Genelabs | General population |
| Turkey | 1 | 2001 | General Biologicals | Children (6 months -15 years) |
| Turkey | 1 | 2002 | Genelabs | General population |
| Turkey | 1 | 2004 | unspecified | Children |
| Turkey | 1 | 2009 | unspecified | General adult population |
| WHO South-East Asia Region (n = 7) | | | | |
| Bangladesh | 1 | 2009 | In-house | General population |
| Bhutan | 1 | 1997 | Abbott | General population and pregnant women |
| India | 1 | 1982, 1992 | unspecified | General population |
| India | 1 | 1999-2000 | Genelabs | General population |
| Indonesia | 1 | 2004 | In-house | Healthy individuals |
| Thailand | 1 | 2000 | unspecified | unspecified |
| Thailand | 1 | 2002 | Euroimmun | Hmong ethnic population |
| WHO Western Pacific Region (n = 14) | | | | |
| China | 1 | 1998 | unspecified | Families in general population |
| China | 1 | 2003-2006 | Wantai | Residents in Jilin province, during annual health exams |
| China | 1 | 2005-2007 | unspecified | General population, swine, and chickens in Sichuan region |
| China | 1 | 2009 | unspecified | People, swine, and chickens in Beijing region |
| China | 1 | 2009 | unspecified | General population |
| China | 1 | 2010 | Wantai | Individuals from four regions and three ethnic groups |

*(Continued)*

**Table 4.** (Continued)

| Country | Number of Sources | Year | Testing Kit Brand | Source Population |
|---|---|---|---|---|
| China | 1 | 2012 | In-house | Healthy people from four ethnic minorities |
| Mongolia | 1 | 2014 | In-house | Healthy individuals |
| Republic of Korea | 1 | 1995 | Genelabs | Volunteers 40–60 years and swine |
| Republic of Korea | 1 | 2003-2004 | Genelabs | General population |
| Singapore | 1 | 1996 | Genelabs | Hospitalized patients with and without liver conditions |
| Taiwan | 1 | 1995 | Diagnostic Biotechnology | Healthy individuals |
| Taiwan | 1 | 2004 | Abbott | Preschool children |
| Taiwan | 1 | 2005 | Diapro | Patients with chronic hepatitis B and chronic hepatitis C |

**HE estimates by WHO regions. WHO South-East Asia Region:** HEV has been reported as highly endemic in several parts of Asia (south, central and southeast Asia) [54]. As a region, SEARO has the highest burden of HEV disease with an incidence rate of 350 per 100,000 [55]. According to the Global Burden of Disease Study in 2017, in SEARO Bangladesh and India reported the highest incident rates of HEV, with incidence rates of 468.08 (CI: 384.58, 566.58) and 389.17 (CI: 327.02, 481.91) per 100,000 population, respectively [56]. Cases in South-East Asia countries are largely attributed to genotype 1 [41].

There have been documented HEV outbreaks in several South-East Asian countries, likely caused by HEV genotype 1: Bangladesh [57,58], India [59–92], Myanmar [93], Nepal [94] and Pakistan [95–100] (Table 5). The first confirmed HEV outbreak in the region occurred in New Delhi, India in 1955 with about 29 000 reported cases and an attack rate 2.05% [60,63]. The majority of HEV outbreaks in South-East Asia have been reported from India with at least 30 reported outbreaks ranging from 150 cases to 29,000 cases reported since 1955 [59–92].

Bangladesh reported two HEV outbreaks [57,58]. One outbreak with more than 4000 cases with a 4% attack rate was reported from Dhaka urban area [57]. Similarly, one HEV outbreak has been reported from Myanmar (Burma) [93].

Although the overall WHO region specific seroprevalence data is not available, the overall seroprevalence in Asia is estimated to be 16% [36]. However, there is wide variation of estimates, ranging from 0% in Mongolia to approximately

**Table 5. Epidemiology of Hepatitis E virus in WHO South-East Asia Region.**

| Country | Human Genotype | Anti-HEV Seropositivity (IgG) | IHME 2017 (No cases) | Reported Outbreaks? | Year (most recent) | Sources |
|---|---|---|---|---|---|---|
| WHO South-East Asia Region | | | | | | |
| Bangladesh | 1,2 | 22.5-60.1% | 672,470 | Yes | 2018 | [101,102] |
| Bhutan | 1,2 | 2.0% | 1,579 | No data | No data | [103] |
| India | 1 | 4.78-17.8% | 5,531,553 | Yes | 2019 | [41,104–106] |
| Indonesia | 1,2,3 | 11.6-52.4% | 665,030 | Yes | 1998 | [107–109] |
| Maldives | No data | No data | 988 | No data | No data | NA |
| Myanmar (Burma) | No data | No data | 134,098 | Yes | 1989 | [93,110] |
| Nepal | 1 | 34.9-43.0% | 95,329 | Yes | 2014 | [40,111] |
| North Korea | No data | No data | No data | No data | No data | NA |
| Sri Lanka | No data | 0.4% | 49,504 | No data | No data | [112] |
| Thailand | 3 | 4.8-22% | 121,933 | Yes | 2011 | [113,114] |
| Timor-Leste | No data | No data | 3,963 | No data | No data | NA |

19% in the United Arab Emirates [36]. Moreover, many countries in South-East Asia, Western Pacific and Eastern Mediterranean region do not have any published estimates of seropositivity or clinical proportion of hepatitis cases attributable to HEV, despite often being classified as endemic (Tables 5–7).

**WHO Western Pacific Region:** Among the countries in Western Pacific region, China was found to report the highest incidence of HEV. According to the Global Burden of Disease Study in 2017, China reported the highest incident rates of HEV, with incidence rates of 380.34 cases per 100,000 population. One of the largest and most prolonged reported outbreaks in the world was reported from Xinjiang, China [41]. This outbreak, with 120 000 suspected cases and an overall attack rate of 3.0% [115], lasted from September 1986–April 1988 [41]. Similarly, several others HEV outbreaks have been documented in China [107,116,117], likely caused by HEV genotype 1. However, endemic cases in Western Pacific Region, including China and Japan, have been largely caused by genotype 3 and 4 [118–120]. Indonesia reported two HEV outbreaks, in East Java [107] and Kalimantan Island [116,117]. One HEV outbreak has been reported from Vietnam [121]. Table 6 shows the current gaps in Western Pacific countries.

**WHO Eastern Mediterranean Region:** In the Eastern Mediterranean region several HEV outbreaks have been reported in Pakistan [95–100], Iraq [149], Egypt [150], Morocco [151,152], Somalia [153,154], and Sudan [48–50,155,156] with no available data for many countries (Table 7). Four HEV outbreaks with attack rates ranging from 10.4% to 20% of the population were reported in Pakistan [95–99]. Similarly, in Iraq an outbreak with more than 250 suspected cases of HEV was reported in 2005 [149].

Pregnant women in Egypt were found to have extremely high seroprevalence, approximately 85%, and very little clinical disease is observed during pregnancy [157]. The epidemiology in Egypt is unique in that most studies find between 50 and 80% seroprevalence, yet very few clinical cases of hepatitis are attributed to HEV [158].

**WHO Africa Region:** HEV is responsible for a large proportion of acute hepatitis outbreaks in African region with Sub Saharan Africa as the acute endemic zone. A relatively recent systematic review [41] found HEV outbreaks in 14 African countries (Table 8): South Africa [158], Kenya [43,44], South Sudan [177], Central African Republic [178–180], Republic of Djibouti [181], Algeria [182–185], Chad [47,48,182–184,186,187], Namibia [8,46], Cameroon [188], Ethiopia [189], and Uganda [51,190–193]. Many of these outbreaks have occurred in refugees or camps for internally displaced populations.

Out of 49 Sub Saharan Africa (SSA) countries, HEV clinical cases has been reported in 25 countries with 20 outbreaks being reported in 9 countries across SSA: Chad, Nigeria, Central African Republic, Ethiopia, Kenya, Uganda, Namibia, Sudan, and Somalia [42]. With more than 10000 suspected HEV cases, Uganda reported the highest number of cases [194]. One study estimates 10–60% of sporadic jaundice cases from anywhere in Africa region to be attributed to hepatitis HEV [168]. However, these are often small, convenience samples and HEV is not usually considered in the differential diagnosis. Case-fatality estimates in the general population during outbreaks in Africa range from 1.5% in Uganda [194] to 23% in Chad [195]. As in Asia, increased risk of severe disease and mortality is seen in infected pregnant women, ranging from 18.8% in Sudan [196] to 65.2% in Uganda [197]. However, children under two years old also have an increased risk of mortality, with approximately 13% case-fatality rate reported from a large, protracted outbreak in among displaced persons in Uganda [191].

Genotype 1 and 2 are most commonly isolated as the cause of HEV from Africa [168]. Across African countries, the total population seroprevalence is estimated to be 22% [36]. However, wide variation is observed with an overall seroprevalence of 0.4% in a random community sample in Nigeria to 100% in displaced persons of Chad [42]. Seroprevalence differs between urban and rural populations, but this is not consistent across countries. In South Africa, seroprevalence is higher in rural areas while in Gabon it is higher in urban areas [168]. However, most sub-Saharan African countries do not have any published estimates of the seroprevalence, clinical burden, or documented outbreaks, despite often being classified as endemic areas (Table 8).

**WHO Europe Region:** There have been many studies that report a substantial increase in locally acquired HEV cases in nearly all European countries in the first decade of 21st century [247–249]. However, it is unclear whether this increase

**Table 6. Epidemiology of Hepatitis E virus in WHO Western Pacific Region.**

| Country | Human Genotype | Anti-HEV Seropositivity (IgG) | IHME 2017 (No cases) | Reported Outbreaks? | Year (most recent) | Source |
|---|---|---|---|---|---|---|
| WHO Western Pacific Region | | | | | | |
| American Samoa (USA) | No data | No data | 154 | No data | No data | NA |
| Australia | 1, 3,4 | 5.9-8% | 24,447 | Yes | 2014 | [122–124] |
| Brunei | No data | No Data | 660 | No data | No data | NA |
| Cambodia | 1,3, 4 | 18.4-76% | 44,677 | Yes | 1994 | [121,125,126] |
| China | 1, 3, 4 | 22.7% | 3,749,848 | Yes | 1986-1988 | [127] |
| Cook Islands Fiji | No data | No data | No data | No data | No data | NA |
| French Polynesia (France) | 3 | No data | No data | 1st case | 2015 | [121] |
| Guam (USA) | No data | No data | 416 | No data | No data | NA |
| Hong Kong SAR (China) | 1 | 18.8% | No data | No data | No data | [128,129] |
| Japan | 3 | 2.9-6.4% | 156,772 | Yes | 2016 | [120,130,119] |
| Kiribati | No data | 6.0% | 368 | No data | No data | [131] |
| Laos | 4 | 41-59.1% | 19,677 | No data | No data | [132,133] |
| Macao SAR (China) | No data | No data | No data | No data | No data | NA |
| Malaysia | No data | 5.9-10.3% | 64927 | No data | No data | [134,135] |
| Marshall Islands | No data | No data | 165 | No data | No data | NA |
| Federated States of Micronesia | No data | No data | 299 | No data | No data | NA |
| Mongolia | 4 | 6.0% | 7,463 | Yes | 2013 | [136,137] |
| Nauru | Na data | No data | No data | No data | No data | NA |
| New Caledonia (France) | 3 | 17.3% | No data | Yes | 2023 | [138] |
| New Zealand | 3 | 4% | 4,877 | No data | No data | [139] |
| Niue | No data | No data | No data | No data | No data | NA |
| Northern Mariana Islands (USA) | No data | No data | 95 | No data | No data | NA |
| Palau | No data | No data | No data | No data | No data | NA |
| Papua New Guinea | No data | 15.2% | 33,779 | No data | No data | [131] |
| Philippines | 3 | 11.8% | 284,396 | No data | No data | [140,141] |
| Pitcairn Islands (UK) | No data | No data | No data | No data | No data | NA |
| Samoa | No data | No data | 602 | No data | No data | NA |
| Singapore | 3 | 14-35%% | 7,457 | Yes | 2016 | [142] |
| Solomon Islands | No data | No data | 2,087 | No data | No data | NA |
| South Korea | 3, 4 | 7.5-17.7% | 66,900 | Yes | 2022 | [143,144] |
| Taiwan | Taiwan genotype | 3.7-26.7% | 38,361 | No data | No data | [145,146] |
| Tokelau | No data | No data | No data | No data | No data | NA |
| Tonga | No data | No data | 298 | No data | No data | NA |
| Tuvalu | No data | No data | No data | No data | No data | NA |
| Vanuatu | No data | No data | 929 | No data | No data | NA |
| Vietnam | 4 | 9% | 222,483 | Yes | 1994 | [121,147,148] |
| Wallis and Futuna (France) | No data | No data | No data | No data | No data | NA |

in cases reflects an increase in transmission or better case detection. Genotype 3 infections are a common cause of acute viral hepatitis in Europe [250,251]. From 2014-2015, France, Germany, and the UK reported more cases of acute hepatitis E than hepatitis A or acute hepatitis B [250]. It has been estimated that there are 68,000 HEV infections in France [252],

**Table 7. Epidemiology of Hepatitis E virus in WHO Eastern Mediterranean Region.**

| Country | Human Genotype | Anti-HEV Seropositivity (IgG) | IHME 2017 (No cases) | Reported Outbreaks? | Year (most recent) | Sources |
|---------|----------------|-------------------------------|----------------------|---------------------|--------------------|---------|
| WHO Eastern Mediterranean Region | | | | | | |
| Afghanistan | 1 | 11-28.4% | 89,016 | No data | No data | [159,160] |
| Bahrain | No data | No data | 2,253 | No data | No data | NA |
| Egypt | 1,3 | 20.21-84% | 186,827 | Yes | 2008 | [157,161] |
| Iran | 1 | 10.95-28.3%% | 127,070 | Yes | 2012 | [161–163] |
| Iraq | No data | 20.3.2% | 88,520 | Yes | 2006 | [164] |
| Jordan | 1 | 30.8% | 20,149 | No data | No data | [165] |
| Kuwait | No data | 98.0% | 6,743 | No data | No data | [166] |
| Lebanon | No data | 21.6% | 14,048 | No data | 2016 | [167] |
| Morocco | 1 | 8.5-10.4% | 62,183 | Yes | 2001 | [41,168] |
| Oman | No data | No data | 7,783 | No data | No data | NA |
| Pakistan | 1, 7 | 10-89.4% | 731,030 | Yes | 2008 | [96,98,169–171] |
| Qatar | 1, 3 | 20.4-32.1% | 4,212 | No data | No data | [172,173] |
| Saudi Arabia | No data | 9.1-15.2% | 77,057 | No data | No data | [161,174] |
| Syria | No data | 31.9% | 34,204 | Yes | 1998 | [175] |
| Tunisia | 1,3 | 5.68-46% | No data | Yes | 1991 | [161,176] |
| Yemen | No data | 10.6% | 55,754 | No data | No data | [161] |

100,000 in the United Kingdom [253] and 300,000 in Germany annually [254]. In a relatively recent semi-structured survey in 30 European countries, it was found that the total number of reported cases of HE for a population of 469 million people has increased from 514 per year in 2005–5,617 in 2015 [255]. Another study estimates 545 hospitalizations and 18 deaths in France per year due to HEV infection [252]. In the Netherlands, HEV infection was found to be the most frequently diagnosed cause of acute viral hepatitis between 2013 and 2015 [256]. In Italy, a small HEV outbreak with five confirmed cases of HEV-4 infections was reported [257].

There have been documented HEV outbreaks in Uzbekistan [258] and Turkmenistan [259], likely caused by HEV genotype 1. Turkmenistan reported a large HEV outbreak, with more than 16 000 cases [259]. However, European outbreaks tend to be small, with only a few cases reported in each (Table 9). The overall seroprevalence of HEV antibodies in Europe is estimated to be 9% [36], yet the variation in the seroprevalence estimates of hepatitis E is noticeable across countries (Table 9). Hepatitis E is hyperendemic in southwest France, with seroprevalence rates of >50% [260], and endemic in northern France, United Kingdom, Belgium, Netherlands, Luxembourg and Germany where 10–30% of individuals have serological evidence of previous HEV exposure [4]. However, adults in Scotland have been found to have a low seroprevalence (<5%) [260]. Children 2–4 years old from Southwest France also had a low seroprevalence of 2% [260].

**WHO Americas Region:** In the Americas, the first HEV outbreak with more than 200 suspected cases and with an overall attack rate of 5%-6% was reported in Mexico in 1986 [328,329]. Another study has reported locally acquired cases of HEV infections from California in the United States [330]. While seroprevalence has been reported from Argentina, Brazil, Bolivia, Chile, Colombia, Peru, Uruguay, Venezuela in the South America [331], there is a paucity of information regarding the clinical presence of HEV in South America (Table 10). The overall seroprevalence in North America is estimated to be 8% [36] (Table 10). However, in the United States, a decreasing seroprevalence of HEV antibodies has been found in recent studies. A study documented unexpectedly high rates, over 20%, of HEV antibodies in blood samples among the general population who were residing in the South, Northwest, Midwest and West during the early 2000s [332]. However, a more recent study shows a decrease to 9% prevalence of anti-HEV IgG [333]. Both studies were completed using samples collected for the National Health and Nutrition Exanimation Survey (NHANES), which is a series

**Table 8.  Epidemiology of Hepatitis E virus in WHO African Region.**

| Country | Human Genotype | Anti-HEV Seropositivity (IgG) | IHME 2017 (No cases) | Reported Outbreaks? | Year (most recent) | Sources |
|---|---|---|---|---|---|---|
| WHO Africa Region | | | | | | |
| Algeria | 1 | 17.05% | 72,659 | Yes | 1987 | [41,185,198] |
| Angola | No data | 40.0% | 119,247 | Yes | 1999 | [199] |
| Benin | 1,2 | 16.19% | 36,388 | Yes | 2011 | [200] |
| Botswana | No data | No data | 5,262 | Yes | 1985 | [201] |
| Burkina Faso | 2,3 | 11.6-39.0% | 72,822 | Yes | 2016 | [202–204] |
| Burundi | No data | 4-14.0% | 38,347 | Yes | 1993 | [205,206] |
| Cape Verde | No data | No data | 1,265 | No data | No data | NA |
| Cameroon | 1,3,4 | 5.8% | 86,446 | Yes | 2017 | [42,188,207] |
| Central African Republic | 1 & 2 | 24.2-79.5% | 20,375 | Yes | 2005 | [168,179] |
| Chad | 1 & 2 | 23.5-75.9% | 56,095 | Yes | 2017 | [168,208] |
| Democratic Republic of the Congo | 3 | 10.4% | 337,787 | Yes | 2011 | [209,210] |
| Djibouti | 1 | 19% | 2,839 | Yes | 1993 | [181] |
| Equatorial Guinea | No data | No data | 5,299 | No data | No data | NA |
| Eritrea | No data | 26.8% | 16,403 | Yes | 2017 | [211] |
| Eswatini | No data | No data | No data | No data | No data | NA |
| Ethiopia | No data | 31-59.0% | 324,288 | Yes | 2015 | [212,213] |
| Gabon | No data | 6.6-14.1% | 5,892 | Yes | 2007 | [212,214] |
| Gambia | No data | 13.7% | No data | Yes | 2012 | [215] |
| Ghana | No data | 12.2-45.3% | 83,733 | Yes | 2017 | [216,217] |
| Guinea | 3 | 2.7% | 40,087 | No data | No data | [218,219] |
| Guinea-Bissau | No data | No data | 6,246 | No data | No data | NA |
| Cote d'Ivoire | No data | 1.5-18.4 | 77,896 | Yes | 2018 | [220] |
| Kenya | No data | 37.8% | 143,213 | Yes | 2012 | [221,44] |
| Lesotho | No data | No data | 4,765 | No data | No data | NA |
| Liberia | No data | No data | 15,312 | No data | No data | NA |
| Libya | No data | 4.49-10% | 12,276 | Yes | 2022 | [222,223] |
| Mali | No data | No data | 71,132 | No data | No data | NA |
| Madagascar | 3 | 14.1% | 72,402 | No data | No data | [224] |
| Malawi | No data | 7.5-16.5% | 43,963 | Yes | 2008 | [225,226] |
| Mauritania | No data | No data | 11,155 | No data | No data | NA |
| Mauritius | No data | No data | 2,233 | No data | No data | NA |
| Mozambique | No data | No data | 81,798 | No data | No data | NA |
| Namibia | 1 & 2 | 25% | 5,800 | Yes | 2019 | [46,168,227] |
| Niger | No data | No data | 79,466 | Yes | 2017 | [228] |
| Nigeria | 1, 2; & 3 | 13.4-94.0%% | 664,456 | Yes | 2017 | [42,168] |
| Rwanda | 3 | 11.9% | 30,948 | No data | No data | [229] |
| Sao Tome and Principe | 3 | No data | 539 | No data | No data | [230] |
| Senegal | 2,3 | 7.4-64.62% | 43,322 | Yes | 2014 | [231–233] |
| Seychelles | No data | No data | 195 | No data | No data | NA |
| Sierra Leone | 3 | 4.0% | 26,690 | No data | No data | [234] |
| Somalia | No data | 46.7% | 50,855 | Yes | 1993 | [235] |
| South Africa | 3 | 10.7-29.5% | 126,341 | Yes | 2013 | [236–238] |
| South Sudan | 1 | 71.0% | No data | Yes | 2023 | [239,240] |
| Sudan | 1 | 5.4-59.0 | 102,000 | Yes | 2014 | [168,241] |

*(Continued)*

**Table 8.** (Continued)

| Country | Human Genotype | Anti-HEV Seroposi-tivity (IgG) | IHME 2017 (No cases) | Reported Outbreaks? | Year (most recent) | Sources |
|---|---|---|---|---|---|---|
| Tanzania | No data | 0.2-8.0% | 102,506 | No data | 2011 | [168,242] |
| Togo | No data | 5.6% | 22,745 | No data | No data | [243] |
| Uganda | 1 & 3 | 47-64.4% | 116,229 | Yes | 2014 | [191,197,244] |
| Zambia | 3 | 8-42.0% | 44,380 | No data | No data | [245,246] |
| Zimbabwe | No data | No data | 39,290 | No data | No data | NA |

of cross-sectional studies designed to be representative of the United States population. In South America, the overall seroprevalence is estimated to be 7%, with ranges from 3.8% in Venezuela to 17.5% in Chile. However, no outbreaks have been reported from South America [10].

**What are the risk factors for HEV infection?**

The qualitative synthesis of reported risk factors across 395 study entries revealed five distinct thematic domains (Fig 2). The Waterborne Pathway emerged as the most prominent driver, led by reports of contaminated water (n = 178) and poor sanitation (n = 88). This was followed by the Zoonotic Pathway, primarily driven by pork product consumption (n = 102). Host-related factors, specifically older age (n = 76) and immunosuppression (n = 42), along with contextual risks like displacement (n = 44), rounded out the evidence landscape. These frequencies reflect the dual nature of HEV as both a sanitation-related and foodborne pathogen, with clinical profiles varying significantly by age and immune status.

**Age.** In Southeast Asia and Africa, HEV is transmitted as a fecal-oral pathogen, with presumably constant environmental risk for exposure. Therefore, it is expected that children would be exposed to and infected at an early age. However, HEV antibody seroprevalence is very low in young children in Southeast Asia (less than 10%) [368]. Prevalence of HEV antibodies increases the most in those ages 15–30 years, leveling off in the 30s [369]. Additionally, most clinical cases of HE are seen in young adults and pregnant women [4,40]. In contrast, about 75–90% of children in endemic areas will have antibodies against hepatitis A virus, which is also transmitted through the fecal-oral route, by 10 years old and there are very few clinical hepatitis A cases [370]. This high antibody prevalence to hepatitis A virus and limited clinical cases indicates nearly ubiquitous, asymptomatic, or mild infection in young children. It is not clear why children in HEV genotype 1 and 2 endemic areas have such a low seroprevalence, and presumably low infection rates. One study from Bangladesh found that seropositive children were less likely to have detectable antibodies 10 years after an HEV infection than adults [368]. Twenty percent (95% CI: 12.0, 28.0) of the participants who were children had no detectable antibodies at follow-up after 10 years [368]. In an outbreak in Chad, children had low rates of disease, but the highest prevalence of IgM antibodies, indicating a current or recent (within the last 6 months) infection [195,208].

**Sanitation and access to clean water.** HEV genotypes 1 and 2 are primarily transmitted through the fecal-oral route, usually by consuming contaminated water. Several HEV outbreaks have been linked to contaminated water supplies, improper water storage, and inadequate chlorination [41,187,46]. Therefore, access to adequate sanitation facilities and clean water are vital to reduce exposure to HEV. In many geographical locations where HEV genotypes 1 and 2 are endemic, the water sources that are available usually include wells, ponds, and rivers. These water sources are used for purposes such as cooking or basic hygiene practices and people may defecate near the water sources leading to contamination. Additionally, refugee and internally displaced person camps, migrant settlements, or low-income housing including slum/squatter communities and informal settlements may not have well-regulated infrastructure, including water and sanitation facilities, including limited connections to piped water and exclusion from regional planning [371,372]. In these areas, individuals must wash their hands often or boil the water that is being consumed before using it, which is not always feasible nor practical.

**Table 9. Epidemiology of Hepatitis E virus in WHO Europe Region.**

| Country | Human Genotype | Anti-HEV Seropositivity (IgG) | IHME 2017 (No cases) | Reported Outbreaks? | Year (most recent) | Sources |
|---|---|---|---|---|---|---|
| WHO Europe Region | | | | | | |
| Albania | No data | 21.1% | 4,732 | No data | 1995 | [261] |
| Andorra | No data | No data | 71 | No data | No data | NA |
| Armenia | No data | No data | 5,624 | No data | No data | NA |
| Austria | 3 | 13.6% | 8,061 | No data | No data | [262] |
| Azerbaijan | No data | 27.5% | 20,589 | No data | No data | [263] |
| Belarus | No data | 7.3% | 16,471 | No data | No data | [264] |
| Belgium | 1, 2,3,4 | 7-15% (Swine) | 13,665 | Yes | 2017 | [265,266] |
| Bosnia & Herzegovina | No data | No data | 5,208 | No data | No data | NA |
| Bulgaria | 1, 3 | 9.0% | 10,389 | Yes | 2015 | [267,268] |
| Croatia | 3 | 1.7-27.9% | 6,436 | No data | 2018 | [269,270] |
| Cyprus | No data | 3.0% | 1,172 | No data | No data | [271] |
| Czech Republic | 1,3 | 5-27.8% | 16,051 | Yes | 2012 | [272,273] |
| Denmark | 1, 3, 4 | 4.1-50.4% | 5,399 | Yes | 2012 | [274–276] |
| Estonia | 1, 3 | 1.96-7.8% | 2,245 | Yes | 2017 | [277] |
| Finland | 1,3 | 5.8-27.6% | 5,186 | Yes | 2008 | [278,279] |
| France | 3 | 3.3-52.5% | 51,887 | Yes | 2016 | [260,280,281] |
| Georgia | No data | No data | 6,718 | No data | No data | NA |
| Germany | 1, 3,4 | 6.8-16.8% | 75,823 | Yes | 2011 | [249,254,282] |
| Greece | 3 | 4.8- 16.5% | 4,664 | Yes | 2015 | [283–285] |
| Hungary | 1,3 | 31.0% | 14,531 | Yes | 2020 | [286,287] |
| Iceland | No data | 2.1% | 332 | No data | No data | [288] |
| Ireland | 3 | 5.3-8.0%% | 4863 | No data | No data | [289,290] |
| Israel | 1, 3,7 | 1.81-10.45% | 7,948 | Yes | 2013 | [291–293] |
| Italy | 3,4 | 0.12-49.0% | 38,063 | Yes | 2011 | [257,294,295] |
| Kazakhstan | | 5.5% | 38,225 | Yes | 1950 | [296] |
| Kosovo | No data | 7.7% | No data | Yes | 1999 | [297] |
| Kyrgyzstan | 1 | 29.9% | 15,412 | Yes | 2019 | [298,299] |
| Latvia | No data | No data | 3,261 | No data | No data | NA |
| Liechtenstein | No data | No data | No data | No data | No data | NA |
| Lithuania | No data | 1.2-43.7% (Animal) | 4,770 | No data | No data | [300] |
| Luxembourg | No data | No data | 552 | No data | No data | NA |
| Malta | No data | No data | 398 | No data | No data | NA |
| Moldova | No data | 24.7-51.1% | 8,558 | No data | | [301] |
| Monaco | No data | No data | No data | No data | No data | NA |
| Montenegro | No data | 6.0% | 1,034 | Yes | 2007 | [302] |
| Netherlands | 3 | 6-27.0% | 9,817 | No data | 2012 | [295,303] |
| North Macedonia | No data | No data | No data | No data | No data | NA |
| Norway | No data | 11.4-30.0% | 5,283 | No data | No data | [304,305] |
| Poland | 3 | 15.9-60.8% | 58,938 | No data | No data | [306,307] |
| Portugal | 3 | 16.3% | 9,216 | Yes | 2022 | [308,309] |
| Romania | 3 | 12-14.9% | 30,395 | No data | No data | [310] |
| Russia | 4 | 1.5-16.7% | 155,929 | Yes | 2012 | [311,312] |
| San Marino | No data | 1.5% | No data | No data | No data | [313] |

*(Continued)*

**Table 9.** (Continued)

| Country | Human Genotype | Anti-HEV Seropositiv-ity (IgG) | IHME 2017 (No cases) | Reported Outbreaks? | Year (most recent) | Sources |
|---|---|---|---|---|---|---|
| Serbia | 3 | 15% | 14,385 | No data | No data | [314] |
| Slovakia | 3 | 7.2-21.5% | 8,349 | No data | No data | [315] |
| Slovenia | 1, 3 | 30.2% (Boar) | 3,093 | No data | No data | [316,317] |
| Spain | 3 | 1.3-7.3% | 36,633 | Yes | 2019 | [295,318–320] |
| Sweden | 1, 2, 3 | 18-30.0% | 10,131 | Yes | 2015 | [321,322] |
| Switzerland | 3,4 | 2.6-4.9% | 7,911 | Yes | 2016 | [323,324] |
| Türkiye | 3 | 12.8-17.3% | 131,051 | No data | No data | [325] |
| Turkmenistan | 1 | 13.0% | 11,182 | Yes | 1994 | [259] |
| Ukraine | No data | No data | 80,079 | No data | No data | NA |
| United Kingdom | 1,3,4 | 21.0% | 66,008 | Yes | 2017 | [326,327] |
| Uzbekistan | No data | 71% | 72,606 | Yes | 2005 | [258] |

Basic hygiene practices such as washing hands and boiling water before consumption leads to lower numbers of HEV infection and lower anti-HEV seroprevalence [373]. Usual water chlorination practices are also effective at eliminating HEV from the drinking water supply [374].

**Cultural factors.** Within HEV endemic areas, there are certain cultural and social practices that may impact the presence of Hepatitis E virus. Next to hygiene and sanitation practices [373], dietary habits as well as animal domestication practices have been documented as risk factors for HEV infection [132]. Additionally, other cultural factors such as alcohol consumption, education, and income level, which are often related to socio-economic status, have also been identified as risk factors for HEV infection.

Families that own pigs and cattle may keep them close to the home for food consumption and resources. Farmers, slaughterhouse personnel and other personnel involved in rearing domestic animals and wild animals have been reported to have higher HEV antibody than those not engaged in these occupations [375]. Consuming products such as raw or undercooked pork, raw pig's blood, and fermented pork sausage significantly increases the risk of HEV infection, especially for genotypes 3 and 4 [373]. However, a study found that households where domesticated cattle and pigs are not kept in close proximity, there is still high seroprevalence of HEV in genotype 1 and 2 areas, possibly due to environmental contamination [132].

At the individual level, alcohol over-consumption has been identified to be another lifestyle risk factor for chronic hepatitis E in Europe. Here, cases are usually caused by genotype 3 [9]. Interestingly, a study in Africa found alcohol consumption to be associated with evidence of a past HEV infection, but not water source or house type [376]. However, the mechanism through which alcohol consumption increases the risk for HEV infection is unclear. One explanation is that excessive alcohol consumption leads to chronic liver disease, which can then lead to an increased risk of symptomatic HEV infection due to the compromised state of the liver, rather than increased risk of exposure to HEV [4]. It is also possible that the foods often consumed with alcohol may be more likely to be undercooked or contaminated with HEV, such as outdoor or grilled foods [376]. Increased alcohol consumption is seen more often in rural areas as well as in families of lower economic status [377]. Studies have also shown that an excessive use of alcohol can lead to more severe HEV infection; likely due to the greater risk for hepatic steatosis or hepatic fibrosis in these patients [4].

## Who are the populations at risk for severe disease?

The synthesis of included studies reveals that while the general population is the most frequently examined group (24.8%, n = 98), a substantial portion of the literature focuses on cohorts at elevated risk for severe clinical outcomes. This includes Maternal and Neonatal populations (12.2%, n = 48), Symptomatic patients (15.9%, n = 63), and Clinical High-Risk groups

**Table 10. Epidemiology of Hepatitis E virus in WHO Americas Region.**

| Country | Human Genotype | Anti-HEV Seropositivity (IgG) | IHME 2017 (No cases) | Reported Outbreaks? | Year (most recent) | Sources |
|---|---|---|---|---|---|---|
| WHO Americas Region | | | | | | |
| Anguilla | No data | No data | No data | No data | No data | NA |
| Antigua and Barbuda | No data | No data | 172 | No data | No data | NA |
| Argentina | 3 | 4.4-35.7% | 24,341 | No data | No data | [331,334] |
| Aruba | No data | 6% | No data | No data | No data | [335] |
| Bahamas | No data | No data | 756 | No data | No data | NA |
| Barbados | No data | No data | 520 | No data | No data | NA |
| Belize | No data | 16.0% | 974 | No data | 1995 | [336] |
| Bermuda | No data | Not detected | 106 | No data | No data | [337] |
| Bolivia | 3 | 6.3-7.3% | 40,987 | No data | NA | [338,339] |
| Brazil | 3 | 1.7-38.0% | 199,499 | Yes | 2008 | [331,340] |
| British Virgin Islands | No data | No data | No data | No data | No data | NA |
| Canada | 3 | 5.9-28.5% | 30,330 | No data | NA | [341,342] |
| Cayman Islands | No data | No data | No data | No data | No data | NA |
| Chile | 3 | 1.4-17.0% | 21,402 | No data | No data | [331,343] |
| Colombia | 3 | 5.9-33.6% | 64,865 | No data | No data | [331,344,345] |
| Costa Rica | 3 | No data | 5,813 | No data | No data | [346] |
| Cuba | 1,2 | 1.4-10% | 19,653 | Yes | 2005 | [347–349] |
| Curaçao | No data | No data | No data | No data | No data | NA |
| Dominica | No data | No data | 137 | No data | No data | NA |
| Dominican Republic | 3 | 4.5-19.0% | 23,791 | No data | No data | [350,351] |
| Ecuador | No data | No data | 53,662 | No data | No data | NA |
| El Salvador | No data | No data | 8,184 | No data | No data | NA |
| French Guiana | No data | 6.4% | No data | No data | No data | [352] |
| Grenada | No data | 3.8% | 225 | No data | No data | [337] |
| Guadeloupe | No data | No data | No data | No data | No data | NA |
| Guatemala | No data | 5.0% | 24,858 | No data | No data | [353] |
| Guyana | No data | No data | 1,726 | No data | NA | NA |
| Haiti | 1 | 10.3-71.0% | 31,478 | Yes | 2005 | [354,355] |
| Honduras | No data | 6.0% | 14,079 | No data | No data | [353] |
| Jamaica | No data | 4.1% | 5,830 | No data | No data | [337] |
| Martinique | No data | No data | No data | Yes | 1858 | [356] |
| Mexico | 1,2,3 | 1.6-40.7% | 173,614 | Yes | 2016 | [347,357,358] |
| Montserrat | Not detected | Not detected | No data | No data | No data | [337] |
| Nicaragua | No data | 4.6-8% | 9,172 | No data | No data | [359] |
| Panama | No data | No data | 5.178 | No data | No data | NA |
| Paraguay | No data | 6.0% | 6,806 | No data | No data | [360] |
| Peru | No data | 6.6-17.1% | 107,893 | No data | No data | [358,361] |
| Puerto Rico | No data | No data | 6,275 | No data | No data | NA |
| Saint Kitts and Nevis | Not detected | Not detected | No data | No data | No data | [337] |
| Saint Lucia | No data | 4.2% | 346 | No data | No data | [337] |
| Saint Vincent and the Grenadines | No data | 3.9% | 234 | No data | No data | [337] |
| Sint Maarten | No data | No data | No data | No data | No data | NA |

*(Continued)*

**Table 10.** (Continued)

| Country | Human Genotype | Anti-HEV Seropositivity (IgG) | IHME 2017 (No cases) | Reported Outbreaks? | Year (most recent) | Sources |
|---|---|---|---|---|---|---|
| Suriname | No data | 3.7% | 1,242 | No data | No data | [362] |
| Trinidad and Tobago | No data | 1ST fatal case | 2,651 | No data | No data | [363] |
| Turks and Caicos | No data | No data | No data | No data | No data | NA |
| United States of America | 3 | 2.4-42.0% | 297,711 | No data | No data | [358,364] |
| Uruguay | 1, 3 | 1.2-2.8% | 2904 | No data | No data | [358,365,366] |
| Venezuela | 1, 3 | 1.3-9.7% | 40,745 | Yes | 2008 | [331,347,367] |

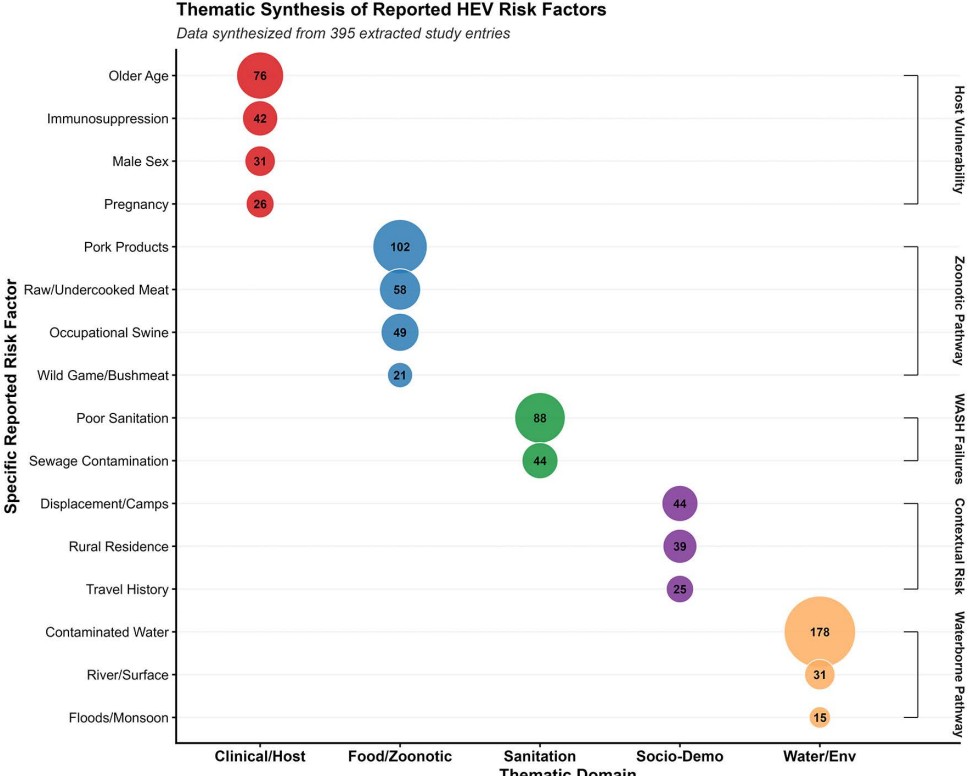

**Fig 2. Thematic synthesis of reported risk factors (N = 395).** A conceptual mapping of identified drivers for HEV infection, categorized into environmental sanitation, zoonotic pathways, and socio-cultural practices.

(9.9%, n = 39), such as the immunocompromised and those with chronic liver disease (CLD). The research focus on these groups is driven by the disproportionate morbidity and mortality they experience, particularly in resource-limited settings.

**Pregnant women and neonates.** Pregnant women and newborns have an alarming mortality rate due to sporadic and outbreak associated HEV infections, largely attributed to genotype 1 or genotype 2 [378]. Mortality in the general population from Hepatitis E is 0.1-4% whereas pregnant women in their third trimester have mortality rates of 10–40% observed in some settings [12–15,379–381]. The risk to the fetus is also high. study from a tertiary hospital in India found an intra-uterine fetal death ratio of 58% [12]. HEV infection during pregnancy increases the risk of low birth weight, the baby being small for gestational age, preterm birth, stillbirth, and intrauterine death [382]. The majority of cases of HEV in

pregnancy occur in resource limited settings in Southeast Asia and sub-Saharan Africa where healthcare infrastructure is weak and there is limited availability of diagnostic testing [378]. Consequently, cases and deaths are not attributed to HEV and the true range of incidence and case fatality rates in pregnancy are not known.

**Immunocompromised patients.** In immunocompromised individuals, HEV infection can become chronic, with HEV RNA remaining detectable for longer than 3 months. Patients at risk for chronic HEV include those who receive immuno-suppressive therapy following solid organ transplantation, stem cell transplantation [383,384], chemotherapy [385], immu-notherapy, or in patients who have concomitant human immunodeficiency virus (HIV) infection [386]. Immunosuppressed patients may present asymptomatic infections or only mildly elevated liver enzymes. However, chronic HEV infection, with persistent viral replication, often progresses to liver fibrosis and cirrhosis [9]. Antibody tests for current or past HEV infec-tion are not recommended for immunosuppressed patients and may remain negative [387].

HEV infection leads to chronic hepatitis in more than 60% of solid organ transplant (SOT) recipients, with one third spontaneously clearing the virus [388]. In addition to fecal-oral transmission, vertical transmission, and transmission from blood products, HEV transmission through solid organ transplant (SOT) in children [389] and adults [390] have also been reported. About ten percent of these patients go on to develop fibrosis and then cirrhosis [388]. Chronically infected transplant patients are usually first treated by having their immunosuppressive treatment reduced for viral clearance [388]. If that is not possible or not sufficient, patients with either chronic or severe acute HEV infection can be treated with ribavirin monotherapy [391,392]. Ribavirin can lead to viral clearance in 78–90% of those treated for 3–6 months [393].

HIV is the most common cause of immune suppression in the world, and much more common than immune suppres-sion from transplantation in areas where the burden of HEV is the greatest. However, most of the research around HEV progression in immune suppressed patients is focused on transplant recipients in Europe [394]. While cases of chronic HEV in HIV positive individuals have been reported, they were all in genotype 3 and 4 endemic areas [386,395,396]. A cross-sectional study from Namibia found that in pregnant women treated with anti-retroviral therapy, the progression and prognosis of infection with HEV is similar to non-HIV infected patients [227]. The 5 co-infected women who were not adherent to antiretroviral therapy had worse outcomes.

**Pre-existing liver disease.** Certain pre-existing health conditions that compromise liver function such as regular overconsumption of alcohol [397,398] and chronic hepatitis B and C infections [4] put individuals at higher risk of clinically apparent HEV infection. Hepatitis E is a potential precipitating factor for developing acute-on-chronic liver failure, leading to rapid decompensation and death [9,387,399,400]. In developing countries, HEV infection with chronic liver disease can lead to high mortality rate, up to 67% within 6 months, although the median is around 30% [401–404]. In Europe, case fatality rates from acute-on-chronic liver failure caused by HEV is reported at 27% [405]. Hepatitis B is a major cause of chronic liver disease worldwide, with a high burden in sub-Saharan Africa. However, there is limited information about HEV-HBV coinfection in that region.

## What are the knowledge gaps related to HEV disease burden estimates?

There is a lack of country-level HEV data in terms of genotypes of acute HEV infections and country-level vital statis-tics which could be explained by the poor surveillance and lack of reporting at the country level. Additionally, incomplete reports of outbreaks as well as the lack of standardized/centralized reporting platforms can cause the overall burden of HE to be vastly underestimated [406]. Many of the countries across WHO regions did not have any reported burden of HEV disease. Similar observations have been made in several systematic reviews and meta-analysis [36,406].

Our global HEV Evidence Maturity Index (EMI) analysis reveals a fractured global landscape where research maturity is frequently inversely proportional to the biological burden of HEV (Fig 3). While EURO (48.1%) and SEARO (54.5%) possess the highest proportions of "Evidence Mature" countries, the WPRO region is dominated by "Absolute Deserts" (Level 1), with 51.4% of nations lacking any primary HEV literature.

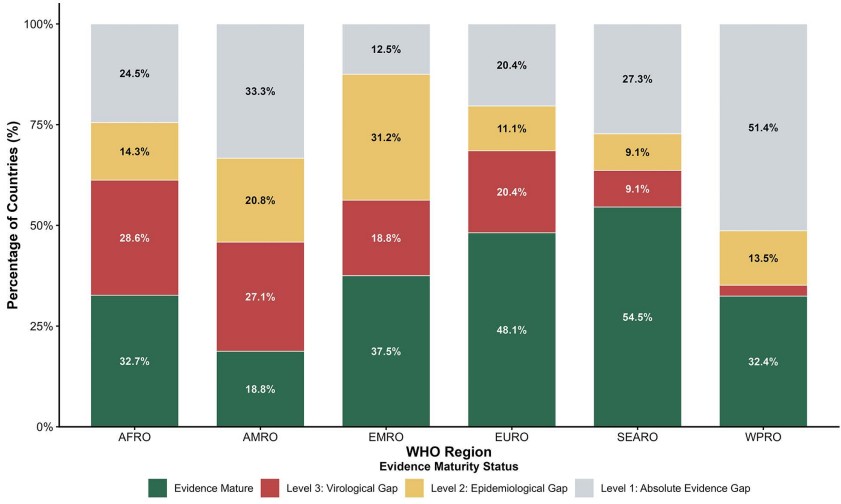

**Fig 3. Global HEV evidence maturity by WHO region.** A visualization of data "maturity" based on the availability of routine surveillance, diagnostic capacity, and peer-reviewed literature density. It highlights the disparity between high-data regions and those with fragmented epidemiological records.

Our review indicates that there is a wide range of variations in the global estimates of mortality and burden of disease caused by HEV. One factor related to these large variations is the lack of a well-established model or sources of data that can be used to estimate the true disease burden caused by HEV. For example, lack of data sources in the LMICs is likely to lead to data disparity and an over representation of vital statistics from developed countries in the widely cited IHME model for HEV mortality. Out of 2,574 national vital statistics reports [32] IHME Global Disease Burden (GBD) 2019 examined, a larger number of reports came from North American and European countries, despite the largest burden of disease from HEV occurring in Southeast Asia and Africa. A similar pattern of data disparity has also been observed in the estimation of GBD 2021 [407].

By plotting the EMI against the $Log_{10}$ estimated annual incidence, we identified significant deviations from the "Balanced Path" of surveillance, highlighting critical "Priority Blind Spots"—geographies where the biological burden drastically outpaces localized knowledge maturity (Fig 4). The AFRO and AMRO regions exhibit the highest discordance, with clusters of high-burden countries—including Mozambique, Mali, and Tanzania in Africa, and Ecuador, Peru, and Guatemala in the Americas—reside deep within these priority blind spot zones.

In GBD 2019 countries such as India, Bangladesh, and Nepal, with a very large burden of disease, did not contribute any sources to these death estimates. Additionally, deaths occurring in outbreaks, particularly in camps for displaced people, are not captured in any national vital statistics as it is often unclear which government is responsible for recording the information. Therefore, it is possible that even the 70,000 deaths per year estimated by Rein et al. [30] may be an underestimate due to lack of reporting. Although their methodology did not allow them to estimate the number of deaths, if we extrapolate the cases to deaths ratio reported by Rein to the 110 million cases per year from Li et al [31], we estimate there are 383,000 or 242,000 deaths, respectively, from HEV per year. Well-designed clinical surveillance and seroprevalence studies based on genotype-specific assays in under-studied areas are needed to accurately estimate the burden of HEV disease.

This molecular need is underscored by the pervasive "Virological Blind Spot" (Level 3) observed across AMRO (27.1%) and AFRO (28.6%), where HEV presence is documented but specific human genotypes remain unknown (Fig 3). Furthermore, over half of the countries in AMRO (71%), WPRO (59%), and AFRO (57%) lack the genotype data required to define regional transmission archetypes (Fig 5).

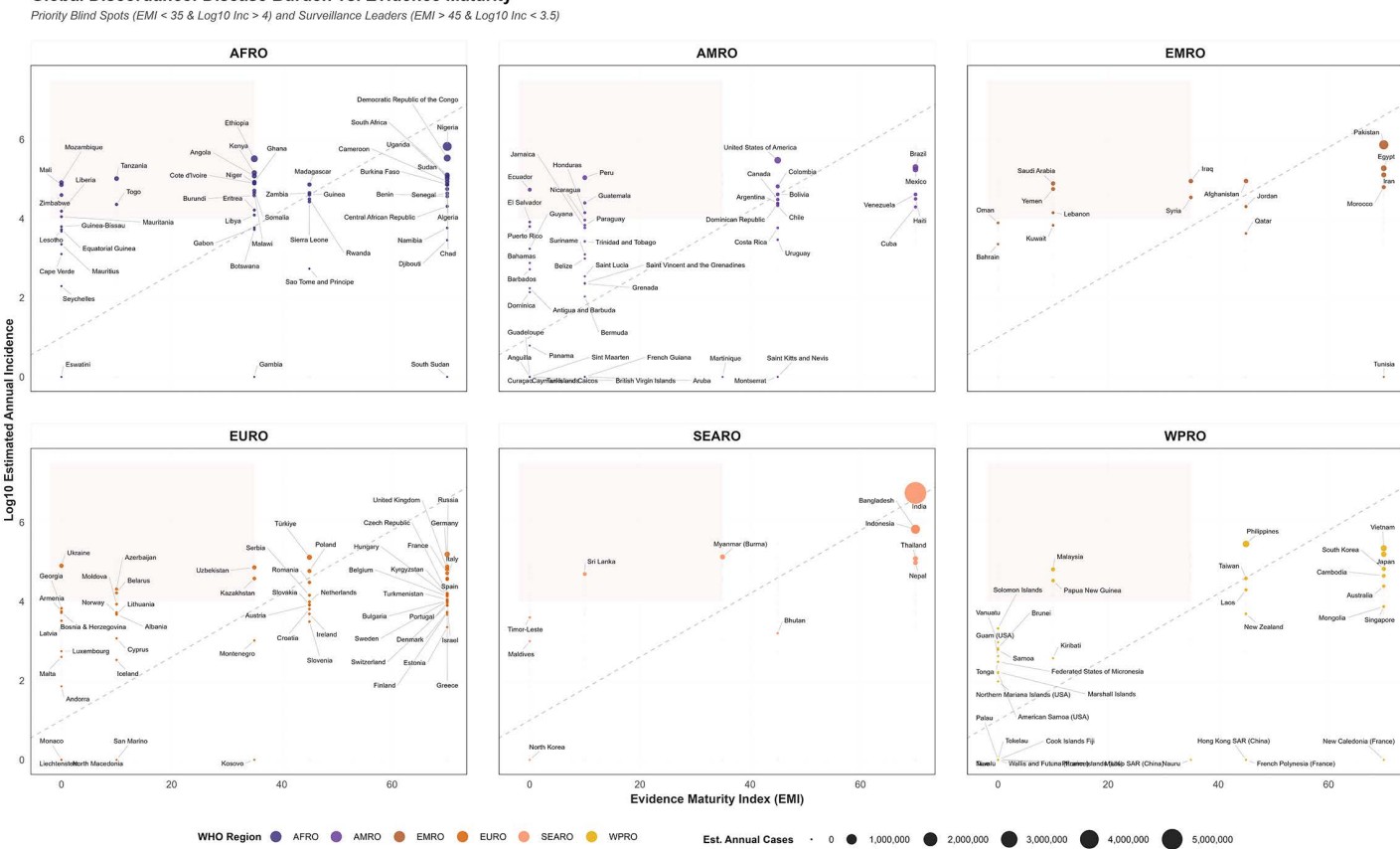

**Fig 4. Global discordance: disease burden vs. evidence maturity.** A comparative analysis illustrating the "Paradox of Data," where regions with the highest estimated HEV incidence often coincide with the lowest levels of evidence maturity and diagnostic infrastructure.

The role of children in transmission of HEV is also not well understood. In Chad, a risk factor for HE was having two or more children under 5 in the household, suggesting that children do play a role in transmission [408]. It is possible that children are not less likely to be infected with HEV but less likely to produce an enduring antibody-generating immune response, thus explaining the low seroprevalence rates seen in children in cross-sectional studies. Current epidemiological and clinical surveillance methods do not identify infections in children well. However, children have not been a focus of the HEV research community due to their low prevalence of IgG antibodies, the major tool to identify people at risk of being infected and thus transmitting infection. This lack of understanding of the role of children in transmission may inhibit using the vaccine to its greatest potential in preventing severe disease. However, the HEV vaccine has not been tested for safety or efficacy in children, and it is unknown if vaccinating children would prevent disease in other age groups.

In terms of risk factors of HEV which vary by genotype, we find age, sanitation and access to clean water and cultural factors generally associated with the global disease burden caused by HEV. Although consumption of products such as raw or undercooked pork, pig's blood or fermented pork sausage have been well documented as risk factors in many studies, there is a lack of clear understanding around alcohol consumption and HE disease burden. This is largely consistent with the findings of other existing studies [409,410]. In addition to current evidence on the occupational exposure

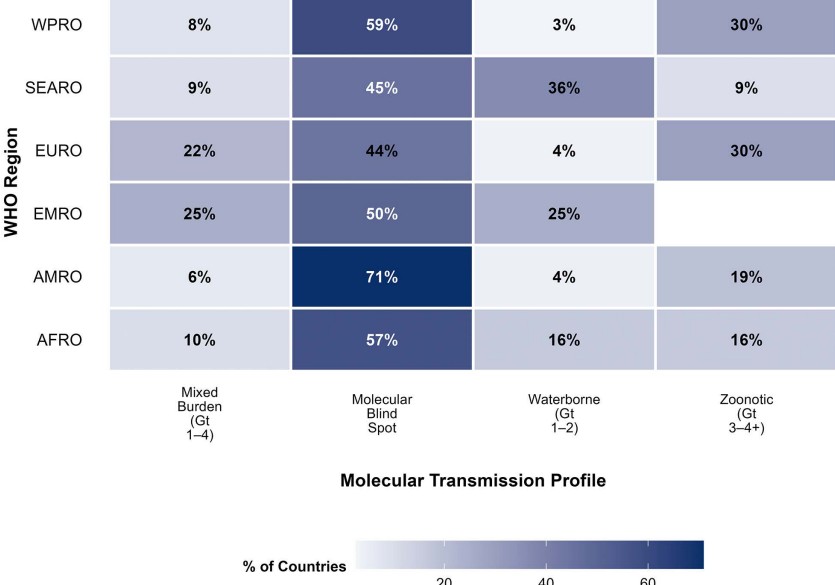

**Fig 5. Regional virological blind spots and transmission profiles.** Map identifying specific geographic areas lacking genotype-specific (molecular) data. It differentiates regions dominated by waterborne Genotypes 1 and 2 from those primarily characterized by zoonotic Genotypes 3 and 4.

to domestic animals, exposure to companion animals including dogs, cats, rabbits and horses has also been reported as potential risk factor of HEV infection [411].

## Discussion

This review presents a comprehensive compilation of current data and associated gaps related to the global disease burden caused by HEV, identifies research priorities and highlights potential solutions to address current knowledge gaps. The underlying factors related to limited understanding of the true global disease burden can be put in three main categories 1) limited public health resources for surveillance, diagnostics, and reporting of cases and deaths in highest risk settings; 2) exclusion of outbreaks from estimates of the burden of disease and 3) unreliable convenience sample derived estimates. In addition, there are important epidemiological knowledge gaps, typically known for other infectious diseases, that are important for developing effective control strategies including: 1) the coexistence of frequent outbreaks and endemic transmission within the same geographic areas; 2) course of infection in children and their role in transmission; 3) the contribution of asymptomatic infections to overall HEV transmission dynamics; and 4) the contribution of zoonotic reservoirs and environmental contamination to sustained HEV transmission in human populations.

HEV genotypes 1 and 2 predominate in areas where water and sanitation systems are poor and public health infra-structure is weak. Thus, in the settings where the risk for water transmitted HEV is high, capacities for surveillance, identi-fication, and reporting of HEV are often most limited. These limited capabilities likely contribute to the lack of attention by global health entities. For example, while WHO Global health sector strategies for HIV, viral hepatitis and sexually trans-mitted infections for the period 2022–2030 acknowledges the importance of viral hepatitis E, its strategic priorities and operational plans focus on eliminating chronic viral hepatitis B and C by 2030 [412].

The findings also indicate that translating seroprevalence (the presence of antibodies against HEV) into the clinical burden of the disease, especially outside of genotype 1 and 2 settings, is a complex issue [101,102]. While seroprevalence data can determine how many people have been exposed to HEV, it doesn't directly correlate to the severity or clinical impact of the infection. This becomes particularly challenging because HEV genotypes vary in their ability to cause severe diseases, and not all infections lead to symptomatic or clinically significant outcomes.

Sporadic cases of endemic hepatitis are often not attributed to HEV, and recognition of HEV as the cause of outbreaks is delayed or never occurs. Therefore, HEV incidence and fatalities per region and country are often unknown in genotype 1 and 2 predominate areas. However, the burden of HEV disease in genotype 3 and 4 predominant areas is over-represented in the global estimates because those countries have the resources and surveillance systems to track and investigate relatively rare events. Therefore, robust surveillance systems, improved facility based diagnostic capabilities and access to point-of-care HEV rapid diagnostic tests in endemic and epidemic settings are required to enhance understanding of burden and offer opportunity for vaccine intervention.

HEV causes substantial, regular outbreaks throughout Southeast Asia and Africa. These outbreaks, particularly in Africa, are often associated with displaced populations. Cases and deaths from HEV outbreaks, especially those associated with displaced people, are not reported in country vital statistics. Additionally, systematic reviews and other global metrics often exclude outbreak-related related data from their estimates [406]. The rationale for exclusion is that outbreaks do not adequately represent the usual burden of disease. However, both large and small outbreaks regularly occur and substantially contribute to the burden of HEV disease. Methods to incorporate outbreak-related cases and deaths are needed to accurately calculate the global burden of disease.

Convenience seroprevalence studies for anti-HEV antibodies are a very useful tool to determine the proportion of a population previously infected with HEV, although generally do not discern genotype. A few sero-surveys were designed specifically to estimate the burden of HEV disease for the general population. A disproportionate number of large, population-based sero-surveys are from countries in Europe, with few or no studies completed in many countries from Southeast Asia and Africa. Many sero-surveys target occupational groups or other special populations and are therefore eliminated from calculations of the burden of disease in the general population. Other sero-surveys use small samples or convenience samples and were not adequately powered to give a precise estimate of the burden of disease, resulting in high variability and large confidence intervals. In addition, other sero-surveys that include children may underestimate the burden of disease since they may be less likely to have lasting antibodies to the infection [369,370]. Therefore, well-designed, population-based sero-surveys would be valuable in the countries of Asia and Africa that may have a high burden.

It is unclear why certain countries in Africa and Asia have a high incidence of endemic disease along with frequent outbreaks [370]. A common paradigm for acute infectious diseases is that the first exposure will cause the host to mount an immune response, and therefore prevent future infections for a time, if not a lifetime. It is unclear why HEV can cause substantial endemic and outbreak disease at the same time and place. Presumably, constant exposure from endemic disease would lead to high enough population immunity to prevent large scale outbreaks. This puzzling pattern of disease implies that the immune response after infection may not follow the usual paradigm. Therefore, control strategies need to be evaluated for their long-term efficacy in preventing both endemic disease and outbreaks.

In general, children are less likely to experience a HEV serological response than older adolescents and adults, though this pattern is not universally observed. Children are under-represented in clinical and epidemiological studies, and little is known about the clinical course and immune response after infection in this population. Furthermore, the role of children in transmission has not been adequately studied. Despite a lower disease burden in children, they may play an important role in transmission. Prospective studies examining the risk of infection and the immune response after infection in children are needed.

To the best of our knowledge this study is one of the very few studies that comprehensively reviews the global evidence base of HEV. However, this review is limited by the selection bias related to the use of bibliographic citations from

3 databases, which has been documented in other studies [413]. While these databases cover the majority of high-impact global health literature, this restriction may have omitted localized or regional clinical reports, potentially underestimating the research activity in specific geographies.

Another limitation is that this review did not incorporate the information from any resources not published in English. This language bias has significant implications for our regional Evidence Maturity Index (EMI), particularly in Latin America (AMRO) and Francophone Africa (AFRO). In these regions, a substantial portion of HEV epidemiology and outbreak data is frequently published in Spanish, Portuguese, or French. Consequently, the "Absolute Deserts" or "Molecular Blind Spots" identified in these regions may partially reflect a lack of English-language reporting rather than an absolute absence of regional research or surveillance activity.

Additionally, this review did not conduct critical appraisal of the included studies. While scoping reviews are designed to map the extent and nature of evidence rather than its quality, the absence of a risk-of-bias assessment remains a key limitation. The consequence of this methodological choice is that the synthesized data is characterized by significant heterogeneity, particularly regarding the reliability of seroprevalence estimates. The use of non-standardized assays and varying case definitions across the 395 included entries may affect the precision of our synthesized burden estimates; therefore, these findings should be interpreted as a thematic map of the evidence landscape rather than a definitive meta-analysis of disease prevalence.

Finally, it must be noted that this review reflects the evidence base available up to December 2024. HEV research is a rapidly accelerating field, and recent studies [414,415] published in early 2025—particularly emerging data from Latin America—are already beginning to fill the "Molecular Blind Spots" identified in our analysis. For instance, while our primary search identified a maximum seroprevalence of 38% in Brazil, recent reports [414,415] indicate that sub-regional IgG markers may reach as high as 59.4%.

The review findings should therefore be viewed as a baseline snapshot of global HEV evidence maturity at the end of 2024.

## Conclusion

The study systematically reviewed global evidence to identify the current estimates of global disease burden, risk factors, and the population at risk for severe HEV infection and analyzed the factors that limit our understanding of HEV epidemiology. The synthesis revealed a fractured evidence base characterized by a severe lack of primary data on HEV incidence and mortality across multiple WHO regions, most notably in the Western Pacific (WPRO) where 51.4% of countries were identified as "Absolute Deserts" (Level 1 maturity). While global mortality estimates vary significantly, the lack of standardized vital statistics in high-burden Low- and Middle-Income Countries (LMICs) is likely a major driver of substantial underestimation. Our analysis identified three core thematic domains of risk that define the current knowledge landscape: environmental waterborne/sanitation deficits (affecting 45.1% of included studies), zoonotic pathways (particularly pork-related exposures), and host-specific vulnerabilities among pregnant women, the immunocompromised, and those with pre-existing liver disease. Furthermore, the discordance mapping successfully identified critical "Priority Blind Spots" in the AFRO and AMRO regions, where the estimated biological burden drastically outpaces localized research maturity. The pervasive absence of genotype data in over half of the countries in these regions constitutes a significant "Molecular Blind Spot," precluding the identification of regional transmission archetypes.

Future research and policy should shift from broad prevalence estimation toward the generation of standardized, comparable datasets. Stakeholders should prioritize:

• The systematic integration of HEV testing into routine clinical diagnostics to mitigate current statistical blindness.

• The expansion of molecular surveillance to resolve unknown transmission archetypes in high-burden regions.

• The development of harmonized global data collection frameworks to capture mortality in high-risk populations, such as those in humanitarian and displaced settings.

Integrating routine HEV diagnostics into national surveillance frameworks offers a pathway to resolving the current statistical invisibility of the disease, providing the high-resolution data necessary to justify and guide equitable global vaccination programs.

## Supporting information

**S1 Checklist. Preferred Reporting Items for Systematic reviews and Meta-Analyses extension for Scoping Reviews (PRISMA-ScR) checklist.** The completed Preferred Reporting Items for Systematic reviews and Meta-Analyses extension for Scoping Reviews (PRISMA-ScR) checklist, confirming adherence to the reporting standards established by Tricco et al. (2018).
(DOCX)

**S1 File. Full electronic search strategies.** Systematic database search strategy for 3 databases.
(DOCX)

**S1 Table. List of included studies and charted data used for evidence synthesis.** A comprehensive spreadsheet detailing the 395 sources analyzed in this review, including study characteristics, geographic locations, and key epidemiological findings.
(XLSX)

**S2 Table. List of excluded studies with reasons for exclusion.** A detailed list of the 70 citations excluded during the full-text screening phase, with specific justifications for exclusion based on the established PCC framework.
(XLSX)

**S3 Table. Charted epidemiological and burden data for 215 countries.** A supplemental data set providing country-level variables, including reported HEV prevalence, case fatality rates, and data availability metrics used to assess global evidence maturity.
(XLSX)

## Author contributions

**Conceptualization:** Md Koushik Ahmed, Alain Labrique, Carl Kirkwood, Kirsten Vannice, Julia Lynch, Brittany L. Kmush.

**Data curation:** Md Koushik Ahmed, Hanna Maroofi, Madeleine Blunt.

**Formal analysis:** Md Koushik Ahmed, Hanna Maroofi, Madeleine Blunt.

**Funding acquisition:** Alain Labrique, Kawsar R. Talaat, Julia Lynch, Brittany L. Kmush.

**Investigation:** Md Koushik Ahmed, Hanna Maroofi, Madeleine Blunt.

**Methodology:** Md Koushik Ahmed, Brittany L. Kmush.

**Project administration:** Brittany L. Kmush.

**Software:** Md Koushik Ahmed.

**Supervision:** Kirsten Vannice, Brittany L. Kmush.

**Validation:** Carl Kirkwood, Kirsten Vannice, Kawsar R. Talaat, Julia Lynch.

**Visualization:** Md Koushik Ahmed, Hanna Maroofi, Madeleine Blunt.

**Writing – original draft:** Md Koushik Ahmed, Hanna Maroofi, Madeleine Blunt, Brittany L. Kmush.

**Writing – review & editing:** Md Koushik Ahmed, Hanna Maroofi, Madeleine Blunt, Alain Labrique, Carl Kirkwood, Kirsten Vannice, Kawsar R. Talaat, Julia Lynch, Brittany L. Kmush.

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
