## [Decision Letter · Decision Letter 0]

10 Nov 2025

Current global estimates, risk factors, and knowledge gaps for Hepatitis E Virus (HEV): A Scoping Review

Dear Dr. Kmush,

Thank you for submitting your manuscript to PLOS Neglected Tropical Diseases. After careful consideration, we feel that it has merit but does not fully meet PLOS Neglected Tropical Diseases's publication criteria as it currently stands. Therefore, we invite you to submit a revised version of the manuscript that addresses the points raised during the review process.

Please submit your revised manuscript within by Jan 09 2026 11:59PM. If you will need more time than this to complete your revisions, please reply to this message or contact the journal office at plosntds@plos.org. Please include the following items when submitting your revised manuscript:

We look forward to receiving your revised manuscript.

Kind regards,

David Safronetz, Ph.D.

Section Editor

David Safronetz

Section Editor

Shaden Kamhawi

co-Editor-in-Chief

Paul Brindley

co-Editor-in-Chief

**Journal Requirements:**

At this stage, the following Authors/Authors require contributions: Md Koushik Ahmed, Hanna Maroofi, Madeleine Blunt, Alain Labrique, Carl Kirkwood, Kirsten Vannice, Kawsar R. Talaat, Julia Lynch, and Brittany L. Kmush. Please ensure that the full contributions of each author are acknowledged in the "Add/Edit/Remove Authors" section of our submission form.

4) We note that your Data Availability Statement is currently as follows: "All relevant data are within the manuscript and its Supporting Information files.". Please confirm at this time whether or not your submission contains all raw data required to replicate the results of your study. Authors must share the “minimal data set” for their submission. PLOS defines the minimal data set to consist of the data required to replicate all study findings reported in the article, as well as related metadata and methods (https://journals.plos.org/plosone/s/data-availability#loc-minimal-data-set-definition).

6) As required by our policy on Data Availability, please ensure your manuscript or supplementary information includes the following:

**Reviewers' Comments:**

Reviewer's Responses to Questions

**Key Review Criteria Required for Acceptance?**

**Methods**

-Are the objectives of the study clearly articulated with a clear testable hypothesis stated?

-Is the study design appropriate to address the stated objectives?

-Is the population clearly described and appropriate for the hypothesis being tested?

-Is the sample size sufficient to ensure adequate power to address the hypothesis being tested?

-Were correct statistical analysis used to support conclusions?

-Are there concerns about ethical or regulatory requirements being met?

Reviewer #1: The abstract describes the answers to the questions, but the inclusion criteria are not identified.

The introduction focuses on items with no direct association with the research problem, and there is no clear rationale for the review; that is, it does not describe why a review with these characteristics is relevant.

The inclusion criteria are unclear. In addition to a description of the PCC framework, more specific criteria for included studies should be included. Exclusion criteria include publications that were not peer-reviewed, but in the search strategy, the authors state that they also reviewed gray literature.

Reviewer #2: Major Revision.

This scoping review addresses an important and underexplored topic: the global burden and epidemiology of Hepatitis E virus (HEV), including genotype distribution, seroprevalence, and outbreak occurrence across WHO regions. The topic is of clear public health relevance and aligns with the current WHO priorities for viral hepatitis elimination.

However, despite the study’s potential and the authors’ effort to follow recognized guidelines, the manuscript requires substantial methodological clarification and better alignment with the Joanna Briggs Institute (JBI) framework and the PRISMA-ScR recommendations. Several aspects of transparency, reproducibility, and reporting standards need improvement before the manuscript can be considered for publication.

1. Alignment with JBI and PRISMA-ScR Frameworks

While the authors mention that the review followed the JBI and PRISMA-ScR recommendations, this adherence is not adequately demonstrated.

- Please explicitly describe each methodological step according to the JBI framework: defining the objectives and review questions, specifying inclusion/exclusion criteria using the PCC structure, explaining data charting, and describing how evidence was collated and summarized.

- Include the PRISMA-ScR checklist as supplementary material, indicating the page or section where each item is addressed.

The two-step search strategy is comprehensive and demonstrates a commendable effort to capture both indexed and gray literature. However, a few aspects compromise reproducibility and methodological transparency:

- The complete search syntaxes should be presented for all databases, not only for PubMed. The current mention of “syntax available in supplementary data” is acceptable but should include a clear summary in the main text.

- The use of Google Scholar for country-specific searches is innovative but must be better justified and methodologically standardized. Please explain how the first 100 results or the first 10 pages, for example, were screened to avoid selection bias.

- Clarify whether the searches were restricted to English only and discuss how this may have influenced geographic representation (language bias).

The PCC structure is appropriate, but its application could be better defined:

- Consider refining the Concept domain: currently, it mixes “factors of risk” and “burden estimates” without clear prioritization. Define whether the main outcome is HEV burden, risk factors, or knowledge gaps.

- Clarify the inclusion of different study designs (cross-sectional, case-control, cohort, surveillance reports) and how these were treated in the synthesis.

- Discuss whether modeling studies or systematic reviews reporting HEV burden estimates were included or excluded, as this affects comprehensiveness.

The use of Rayyan for citation management and double-blind screening is a strength and aligns with good review practices. However:

- Specify how inter-reviewer agreement was measured (e.g., Cohen’s kappa) and what steps were taken to resolve discrepancies.

- The manuscript mentions the use of a priori structured matrices for data extraction. Please describe their structure (variables collected, pilot testing, consistency checks).

- Clarify whether data extraction was duplicated independently or performed by one reviewer and checked by another.

The manuscript describes the extraction of key information (authors, year, genotype, seropositivity, outbreak data), but the analytical strategy remains vague:

- Scoping reviews do not usually perform meta-analysis, but they do require a clear narrative synthesis framework — e.g., grouping by region, study type, or outcome category.

- Describe how findings were summarized (frequency counts, geographical mapping, identification of gaps).

- Consider adding quantitative mapping (tables or figures) showing the distribution of included studies per WHO region, genotypes, and main outcomes.

Finally, the objective is clear, but there is an absence of an explicitly formulated hypothesis.

**Results**

-Does the analysis presented match the analysis plan?

-Are the results clearly and completely presented?

-Are the figures (Tables, Images) of sufficient quality for clarity?

Reviewer #1: The results are consistent with the methodology and are clearly presented; however, the previous comments should be taken into account. The tables are adequate and provide sufficient information.

Reviewer #2: Major Revision.

The authors conducted an extensive review of the international literature on the global epidemiology of hepatitis E. However, the data included in the manuscript are somewhat outdated, and it would be advisable to insert comments in the discussion section to reflect more recent evidence.

For example, the search period for the studies included in this review extended up to December 2024. Nevertheless, recently published data from Latin America, particularly from Brazil, indicate that the prevalence of hepatitis E varies widely. The overall prevalence of the IgG marker in Brazil ranged from 0.5% in the North region to 59.4% in the South region (Mariz et al., 2025). However, in the present review, the authors present a prevalence ranging from 1.7 to 38%, somewhat divergent from what is available in the literature. Similar updated findings have been reported in several other parts of the world.

It is therefore strongly recommended that the authors update the references and include recently published and relevant studies on the hepatitis E virus, such as:

Magri, M. C.; Manchiero, C.; Dantas, B. P.; Bernardo, W. M.; Abdala, E.; Tengan, F. M. Prevalence of hepatitis E in Latin America and the Caribbean: A systematic review and meta-analysis. Public Health 2025, 244, 105745.

Songtanin, B.; Molehin, A. J.; Brittan, K.; Manatsathit, W.; Nugent, K. Hepatitis E Virus Infections: Epidemiology, Genetic Diversity, and Clinical Considerations. Viruses 2023, 15, 1389.

Mariz, C. A., de Araújo, L. R. M. G., & Lopes, E. P. (2025). Hepatitis E Virus Infection in Brazil: A Scoping Review of Epidemiological Features. Pathogens (Basel, Switzerland), 14(9), 895. https://doi.org/10.3390/pathogens14090895

In addition, the tables should be revised to ensure that they are presented in an appropriate format and that references are clearly indicated when the author is cited.

**Conclusions**

-Are the conclusions supported by the data presented?

-Are the limitations of analysis clearly described?

-Do the authors discuss how these data can be helpful to advance our understanding of the topic under study?

-Is public health relevance addressed?

Reviewer #1: The conclusions are consistent with the research questions and the objective of the study. The discussion clearly describes the limitations, scope, and significance of the study, as well as the relevance of the results in the context of public health.

Reviewer #2: Limitations: Adequate transparency but requires a more analytical and updated discussion of the consequences of methodological restrictions.

Excerpt analyzed:

“...this review is limited by the selection bias related to the use of bibliographic citations from 3 databases... we did not incorporate the information from any resources not published in English... Additionally, we did not conduct critical appraisal of the included studies.”

Evaluation:

The limitations are clearly identified and reflect transparency in reporting, which is commendable. The authors appropriately acknowledge potential selection bias due to the restriction to three major databases and to English-only publications. However, the discussion of limitations is too brief and lacks depth in terms of methodological implications.

Points for improvement:

The authors should quantify or qualify the impact of these limitations. For instance, how significant might the exclusion of non-English studies be, particularly in regions such as Latin America, Africa, and Asia, where HEV epidemiology is less represented in English-language literature?

The absence of a critical appraisal (risk of bias assessment) should be further justified. While scoping reviews do not always require formal quality assessment, it is advisable to acknowledge the potential heterogeneity and uneven quality of the included studies and discuss how this may affect the reliability of synthesized evidence.

The authors could also mention the temporal limitation of the search (until December 2024) and note that recent studies have since been published, some of which might alter the global prevalence estimates, including new data from Latin America (e.g., Mariz et al., 2025; Magri et al., 2025).

Conclusion: Conceptually solid and aligned with the study’s aims, but requires refinement in structure and specificity to enhance analytical depth and scientific rigor.

Excerpt analyzed:

“The study systematically reviewed global evidence... found a severe lack of data on HEV incidence and mortality... highlighted potential risk factors... stakeholders should prioritize routine testing and vaccination strategies.”

Evaluation:

The conclusion is coherent and consistent with the objectives of the study. It successfully summarizes the key findings and emphasizes the public health relevance of HEV.

Positive aspects:

The conclusion connects findings to policy and public health implications, particularly the need for equitable vaccination strategies.

It underscores the global neglect of HEV and the variability of existing data, an important contribution to the literature.

Points needing improvement:

The conclusion could differentiate better between what was found and what is recommended. Currently, both are intertwined, reducing the analytical clarity. For instance, the statement “our review highlighted potential risk factors which need to be well understood for future policies” should specify which risk factors (e.g., zoonotic exposure, sanitation deficits, immunosuppression).

The text could also benefit from greater precision and cautious language regarding causality, avoiding phrases that imply direct evidence (e.g., “need to be well understood”) when the review primarily identifies knowledge gaps.

The paragraph might end with a forward-looking sentence emphasizing the need for standardized global data collection frameworks, harmonized surveillance, and integration of HEV testing in routine diagnostics.

Suggested refinement (example excerpt):

“Future research should prioritize the generation of standardized and comparable data across regions, including the systematic integration of HEV testing in routine surveillance. Strengthening global datasets will be critical for guiding vaccination strategies and reducing the burden of this neglected, yet vaccine-preventable, disease.”

**Editorial and Data Presentation Modifications?**

Reviewer #1: In the methodology, the authors should clarify the inclusion criteria and in the introduction they should clearly describe why this review should be conducted.

Reviewer #2: Major Revision.

**Summary and General Comments**

Reviewer #1: The review could contribute to the knowledge on the global epidemiology of HEV, but I have some concerns about its validity and reliability.

Reviewer #2: This manuscript addresses an important and often neglected topic in global health — the epidemiology and disease burden of Hepatitis E virus (HEV). The authors have undertaken a comprehensive scoping review that compiles and synthesizes the global evidence base on HEV prevalence, risk factors, and populations at risk for severe infection. The topic is highly relevant and timely, considering the limited attention HEV receives in public health agendas and the lack of reliable global data on incidence and mortality. The review has the potential to make a valuable contribution to the literature and to inform future research and policy priorities.

However, before the manuscript can be considered for publication, substantial revisions are needed to improve analytical depth, methodological transparency, and interpretative precision. The current version would benefit from a more critical discussion of its methodological limitations, particularly regarding language restriction (English-only studies), database coverage, and the absence of critical appraisal of included studies. These aspects could introduce systematic bias and should be more explicitly addressed in the Limitations section.

In the Conclusion, the manuscript successfully summarizes the main findings and highlights policy implications. Nonetheless, it should more clearly differentiate between observed results and recommended actions, providing greater specificity about the key risk factors identified and the nature of evidence supporting each one. Strengthening the analytical clarity and including a forward-looking perspective emphasizing the need for standardized global HEV surveillance would increase the manuscript’s impact and policy relevance.

In summary, this is a relevant and promising manuscript that addresses an important gap in the global understanding of HEV. I recommend major revision to address the issues noted above, refine the methodological and interpretative sections, and ensure the conclusions are well supported by the evidence synthesized. After these revisions, the study would be a strong candidate for publication.

PLOS authors have the option to publish the peer review history of their article (what does this mean? ). If published, this will include your full peer review and any attached files.

**Do you want your identity to be public for this peer review?** For information about this choice, including consent withdrawal, please see our Privacy Policy .

Reviewer #1: No

Reviewer #2: No

**Figure resubmission:**
---

## [Decision Letter · Decision Letter 1]

28 Jan 2026

Dear Dr. Kmush,

We are pleased to inform you that your manuscript 'Current global estimates, risk factors, and knowledge gaps for Hepatitis E Virus (HEV): A Scoping Review' has been provisionally accepted for publication in PLOS Neglected Tropical Diseases.

Best regards,

David Safronetz, Ph.D.

Section Editor

David Safronetz

Section Editor

Shaden Kamhawi

co-Editor-in-Chief

Paul Brindley

co-Editor-in-Chief

Reviewer's Responses to Questions

**Key Review Criteria Required for Acceptance?**

**Methods**

-Are the objectives of the study clearly articulated with a clear testable hypothesis stated?

-Is the study design appropriate to address the stated objectives?

-Is the population clearly described and appropriate for the hypothesis being tested?

-Is the sample size sufficient to ensure adequate power to address the hypothesis being tested?

-Were correct statistical analysis used to support conclusions?

-Are there concerns about ethical or regulatory requirements being met?

Reviewer #1: The authors describe each research question for developing the review, and the methods used to answer them are consistent. The description of each step in the development process is detailed and sufficient to replicate the results.

Reviewer #2: Accept

**Results**

-Does the analysis presented match the analysis plan?

-Are the results clearly and completely presented?

-Are the figures (Tables, Images) of sufficient quality for clarity?

Reviewer #1: The presentation of results is clear. The distribution by WHO regions is innovative and provides a broader view of the disease burden, while also significantly contributing to answering one of the research questions posed by the authors. The work is original and rigorous, and its strength lies in describing the burden of the disease by region and in analyzing the risk factors in sufficient depth.

Reviewer #2: The results are reported in a clear, coherent, and sufficiently comprehensive manner, allowing an adequate understanding of the study findings. Overall, the presentation of the results meets the standards expected for publication. However, the quality and resolution of the figures (including tables and images) require improvement. Enhancing the sharpness, readability, and overall visual quality of the figures is necessary to ensure clarity and to meet the journal’s publication requirements. Once these issues related to figure quality are adequately addressed, the manuscript will be suitable for publication from the perspective of results presentation.

**Conclusions**

-Are the conclusions supported by the data presented?

-Are the limitations of analysis clearly described?

-Do the authors discuss how these data can be helpful to advance our understanding of the topic under study?

-Is public health relevance addressed?

Reviewer #1: The conclusions are appropriate and directly related to the review's results. Overall, the review fulfills its objective of defining the scope of the current literature, and the results are helpful in identifying the main knowledge gaps, which encourage the reader to generate new research questions about HEV.

Reviewer #2: The authors have satisfactorily addressed all issues raised in the previous round of review.

**Editorial and Data Presentation Modifications?**

Reviewer #1: I suggest maintaining a third-person perspective throughout the discussion section, and adding references on lines 791-793, 793-795, and 825-829.

The abbreviation for HIV is not defined when it first appears in the text.

Reviewer #2: Accept after minor figure adjustments.

**Summary and General Comments**

Reviewer #1: The article seeks to synthesize evidence on HEV and demonstrate the scope of recent literature. In this version, the authors significantly improve the manuscript's presentation by defining the research problem and its resolution through the scoping review. The results are consistent with this methodology, and the current limitations of the published literature are included.

Regarding the concerns raised in the previous review, the authors have addressed each of the comments.

Reviewer #2: The authors are to be commended for the substantial improvements made to the manuscript in response to the reviewers’ comments. The revisions have strengthened the overall clarity, coherence, and scientific rigor of the study. The manuscript addresses a relevant and timely topic and contributes meaningful insights to the existing literature. No concerns related to research ethics, publication ethics, or dual publication were identified. Overall, the authors have adequately addressed all issues raised in previous review rounds, and the manuscript is now suitable for publication.

PLOS authors have the option to publish the peer review history of their article (what does this mean? ). If published, this will include your full peer review and any attached files.

**Do you want your identity to be public for this peer review?** For information about this choice, including consent withdrawal, please see our Privacy Policy .

Reviewer #1: No

Reviewer #2: No

---

## [Editor Report · Acceptance letter]

Dear Dr. Kmush,

We are delighted to inform you that your manuscript, "Current global estimates, risk factors, and knowledge gaps for Hepatitis E Virus (HEV): A Scoping Review," has been formally accepted for publication in PLOS Neglected Tropical Diseases.

Best regards,

Shaden Kamhawi

co-Editor-in-Chief

Paul Brindley

co-Editor-in-Chief
